# Exact expressions for double descent and implicit regularization via surrogate random design

**Michał Dereziński**
Department of Statistics
University of California, Berkeley
mderezin@berkeley.edu

**Feynman Liang**
Department of Statistics
University of California, Berkeley
feynman@berkeley.edu

**Michael W. Mahoney**
ICSI and Department of Statistics
University of California, Berkeley
mmahoney@stat.berkeley.edu

## Abstract

Double descent refers to the phase transition that is exhibited by the generalization error of unregularized learning models when varying the ratio between the number of parameters and the number of training samples. The recent success of highly over-parameterized machine learning models such as deep neural networks has motivated a theoretical analysis of the double descent phenomenon in classical models such as linear regression which can also generalize well in the over-parameterized regime. We provide the first exact non-asymptotic expressions for double descent of the minimum norm linear estimator. Our approach involves constructing a special determinantal point process which we call surrogate random design, to replace the standard i.i.d. design of the training sample. This surrogate design admits exact expressions for the mean squared error of the estimator while preserving the key properties of the standard design. We also establish an exact implicit regularization result for over-parameterized training samples. In particular, we show that, for the surrogate design, the implicit bias of the unregularized minimum norm estimator precisely corresponds to solving a ridge-regularized least squares problem on the population distribution. In our analysis we introduce a new mathematical tool of independent interest: the class of random matrices for which determinant commutes with expectation.

## 1 Introduction

Classical statistical learning theory asserts that to achieve generalization one must use training sample size that sufficiently exceeds the complexity of the learning model, where the latter is typically represented by the number of parameters [or some related structural parameter; see FHT01]. In particular, this seems to suggest the conventional wisdom that one should not use models that fit the training data exactly. However, modern machine learning practice often seems to go against this intuition, using models with so many parameters that the training data can be perfectly interpolated, in which case the training error vanishes. It has been shown that models such as deep neural networks, as well as certain so-called interpolating kernels and decision trees, can generalize well in this regime. In particular, [BHMM19] empirically demonstrated a phase transition in generalization performance of learning models which occurs at an *interpolation thershold*, i.e., a point where training error goes to zero (as one varies the ratio between the model complexity and the sample size). Moving away from this threshold in either direction tends to reduce the generalization error, leading to the so-called *double descent* curve.

To understand this surprising phenomenon, in perhaps the simplest possible setting, we study it in the context of linear or least squares regression. Consider a full rank $n \times d$ data matrix $\mathbf{X}$ and a vector $\mathbf{y}$ of responses corresponding to each of the $n$ data points (the rows of $\mathbf{X}$), where we wish to find the best linear model $\mathbf{Xw} \approx \mathbf{y}$, parameterized by a $d$-dimensional vector $\mathbf{w}$. The simplest example of an estimator that has been shown to exhibit the double descent phenomenon [BHX19] is the Moore-Penrose estimator, $\widehat{\mathbf{w}} = \mathbf{X}^\dagger \mathbf{y}$: in the so-called over-determined regime, i.e., when $n > d$, it corresponds to the least squares solution, i.e., $\mathrm{argmin}_\mathbf{w} \|\mathbf{Xw} - \mathbf{y}\|^2$; and in the under-determined regime (also known as over-parameterized or interpolating), i.e., when $n < d$, it corresponds to the minimum norm solution to the linear system $\mathbf{Xw} = \mathbf{y}$. Given the ubiquity of linear regression and the Moore-Penrose solution, e.g., in kernel-based machine learning, studying the performance of this estimator can shed some light on the effects of over-parameterization/interpolation in machine learning more generally. Of particular interest are results that are exact (i.e., not upper/lower bounds) and non-asymptotic (i.e., for large but still finite $n$ and $d$).

We build on methods from Randomized Numerical Linear Algebra (RandNLA) in order to obtain *exact non-asymptotic expressions* for the mean squared error (MSE) of the Moore-Penrose estimator (see Theorem 1). This provides a precise characterization of the double descent phenomenon for the linear regression problem. In obtaining these results, we are able to provide precise formulas for the *implicit regularization* induced by minimum norm solutions of under-determined training samples, relating it to classical ridge regularization (see Theorem 2). To obtain our precise results, we use a somewhat non-standard random design, based on a specially chosen determinantal point process (DPP), which we term surrogate random design. DPPs are a family of non-i.i.d. sampling distributions which are typically used to induce diversity in the produced samples [KT12]. Our aim in using a DPP as a surrogate design is very different: namely, to make certain quantities (such as the MSE) analytically tractable, while accurately *preserving* the underlying properties of the original data distribution. This strategy might seem counter-intuitive since DPPs are typically found most useful when they *differ* from the data distribution. However, we show both theoretically (Theorem 3) and empirically (Section 5), that for many commonly studied data distributions, such as multivariate Gaussians, our DPP-based surrogate design accurately preserves the key properties of the standard i.i.d. design (such as the MSE), and even matches it exactly in the high-dimensional asymptotic limit. In our analysis of the surrogate design, we introduce the concept of *determinant preserving random matrices* (Section 4), a class of random matrices for which determinant commutes with expectation, which should be of independent interest.

## 1.1 Main results: double descent and implicit regularization

As the performance metric in our analysis, we use the *mean squared error* (MSE), defined as $\mathrm{MSE}[\widehat{\mathbf{w}}] = \mathbb{E}\big[\|\widehat{\mathbf{w}} - \mathbf{w}^*\|^2\big]$, where $\mathbf{w}^*$ is a fixed underlying linear model of the responses. In analyzing the MSE, we make the following standard assumption that the response noise is homoscedastic.

**Assumption 1 (Homoscedastic noise)** *The noise $\xi = y(\mathbf{x}) - \mathbf{x}^\top \mathbf{w}^*$ has mean $0$ and variance $\sigma^2$.*

Our main result provides an exact expression for the MSE of the Moore-Penrose estimator under our surrogate design denoted $\bar{\mathbf{X}} \sim S_\mu^n$, where $\mu$ is the $d$-variate distribution of the row vector $\mathbf{x}^\top$ and $n$ is the sample size. This surrogate is used in place of the standard $n \times d$ random design $\mathbf{X} \sim \mu^n$, where $n$ data points (the rows of $\mathbf{X}$) are sampled independently from $\mu$. We form the surrogate by constructing a determinantal point process with $\mu$ as the background measure, so that $S_\mu^n(\mathbf{X}) \propto \mathrm{pdet}(\mathbf{XX}^\top)\mu(\mathbf{X})$, where $\mathrm{pdet}(\cdot)$ denotes the pseudo-determinant (details in Section 3). Unlike for the standard design, our MSE formula is fully expressible as a function of the covariance matrix $\boldsymbol{\Sigma}_\mu = \mathbb{E}_\mu[\mathbf{xx}^\top]$. To state our main result, we need an additional minor assumption on $\mu$ which is satisfied by most standard continuous distributions (e.g., multivariate Gaussians).

**Assumption 2 (General position)** *For $1 \leq n \leq d$, if $\mathbf{X} \sim \mu^n$, then $\mathrm{rank}(\mathbf{X}) = n$ almost surely.*

Under Assumptions 1 and 2, we can establish our first main result, stated as the following theorem, where we use $\mathbf{X}^\dagger$ to denote the Moore-Penrose inverse of $\mathbf{X}$.

**Theorem 1 (Exact non-asymptotic MSE)** *If the response noise is homoscedastic (Assumption 1) and $\mu$ is in general position (Assumption 2), then for any $\mathbf{w} \in \mathbb{R}^d$, $\bar{\mathbf{X}} \sim S_\mu^n$ (Definition 3), and $\bar{y}_i = y(\bar{\mathbf{x}}_i)$,*

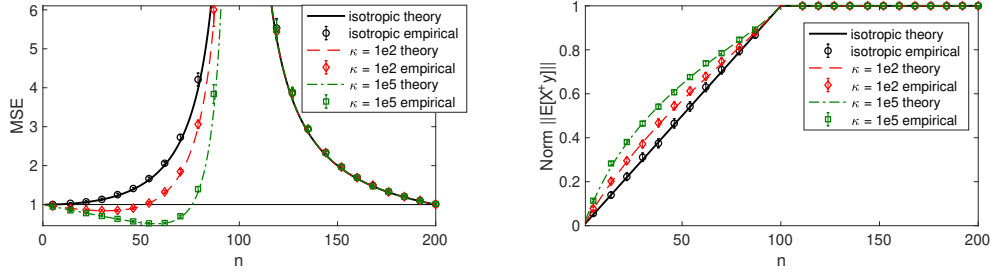

(a) Surrogate MSE expressions (Theorem 1) closely match numerical estimates even for non-isotropic features. Eigenvalue decay leads to a steeper descent curve in the under-determined regime ($n < d$).

(b) The mean of the estimator $\mathbf{X}^\dagger \mathbf{y}$ exhibits shrinkage which closely matches the shrinkage of a ridge-regularized least squares optimum (theory lines), as characterized by Theorem 2.

Figure 1: Illustration of the main results for $d = 100$ and $\mu = \mathcal{N}(\mathbf{0}, \boldsymbol{\Sigma})$ where $\boldsymbol{\Sigma}$ is diagonal with eigenvalues decaying exponentially and scaled so that $\mathrm{tr}(\boldsymbol{\Sigma}^{-1}) = d$. We use our surrogate formulas to plot (a) the MSE (Theorem 1) and (b) the norm of the expectation (Theorem 2) of the Moore-Penrose estimator (*theory* lines), accompanied by the empirical estimates based on the standard i.i.d. design (error bars are three times the standard error of the mean). We consider three different condition numbers $\kappa$ of $\boldsymbol{\Sigma}$, with *isotropic* corresponding to $\kappa = 1$, i.e., $\boldsymbol{\Sigma} = \mathbf{I}$. We use $\sigma^2 = 1$ and $\mathbf{w}^* = \frac{1}{\sqrt{d}}\mathbf{1}$.

$$
\mathrm{MSE}\big[\bar{\mathbf{X}}^\dagger \bar{\mathbf{y}}\big] = \begin{cases} \sigma^2 \, \mathrm{tr}\big((\boldsymbol{\Sigma}_\mu + \lambda_n \mathbf{I})^{-1}\big) \cdot \frac{1 - \alpha_n}{d - n} \;+\; \frac{\mathbf{w}^{*\top}(\boldsymbol{\Sigma}_\mu + \lambda_n \mathbf{I})^{-1}\mathbf{w}^*}{\mathrm{tr}((\boldsymbol{\Sigma}_\mu + \lambda_n \mathbf{I})^{-1})} \cdot (d - n), & \text{for } n < d, \\ \sigma^2 \, \mathrm{tr}(\boldsymbol{\Sigma}_\mu^{-1}), & \text{for } n = d, \\ \sigma^2 \, \mathrm{tr}(\boldsymbol{\Sigma}_\mu^{-1}) \cdot \frac{1 - \beta_n}{n - d}, & \text{for } n > d, \end{cases}
$$

with $\lambda_n \geq 0$ defined by $n = \mathrm{tr}(\boldsymbol{\Sigma}_\mu(\boldsymbol{\Sigma}_\mu + \lambda_n \mathbf{I})^{-1})$, $\alpha_n = \det(\boldsymbol{\Sigma}_\mu(\boldsymbol{\Sigma}_\mu + \lambda_n \mathbf{I})^{-1})$ and $\beta_n = \mathrm{e}^{d - n}$.

**Definition 1** *We will use* $\mathcal{M} = \mathcal{M}(\boldsymbol{\Sigma}_\mu, \mathbf{w}^*, \sigma^2, n)$ *to denote the above expressions for* $\mathrm{MSE}\big[\bar{\mathbf{X}}^\dagger \bar{\mathbf{y}}\big]$.

Proof of Theorem 1 is given in Appendix C. For illustration, we plot the MSE expressions in Figure 1a, comparing them with empirical estimates of the true MSE under the i.i.d. design for a multivariate Gaussian distribution $\mu = \mathcal{N}(\mathbf{0}, \boldsymbol{\Sigma})$ with several different covariance matrices $\boldsymbol{\Sigma}$. We keep the number of features $d$ fixed to 100 and vary the number of samples $n$, observing a double descent peak at $n = d$. We observe that our theory aligns well with the empirical estimates, whereas previously, no such theory was available except for special cases such as $\boldsymbol{\Sigma} = \mathbf{I}$ (more details in Theorem 3 and Section 5). The plots show that varying the spectral decay of $\boldsymbol{\Sigma}$ has a significant effect on the shape of the curve in the under-determined regime. We use the horizontal line to denote the MSE of the null estimator $\mathrm{MSE}[\mathbf{0}] = \|\mathbf{w}^*\|^2 = 1$. When the eigenvalues of $\boldsymbol{\Sigma}$ decay rapidly, then the Moore-Penrose estimator suffers less error than the null estimator for some values of $n < d$, and the curve exhibits a local optimum in this regime.

One important aspect of Theorem 1 comes from the relationship between $n$ and the parameter $\lambda_n$, which together satisfy $n = \mathrm{tr}(\boldsymbol{\Sigma}_\mu(\boldsymbol{\Sigma}_\mu + \lambda_n \mathbf{I})^{-1})$. This expression is precisely the classical notion of *effective dimension* for ridge regression regularized with $\lambda_n$ [AM15], and it arises here even though there is no explicit ridge regularization in the problem being considered in Theorem 1. The global solution to the ridge regression task (i.e., $\ell_2$-regularized least squares) with parameter $\lambda$ is defined as:

$$
\underset{\mathbf{w}}{\mathrm{argmin}} \left\{ \mathbb{E}_{\mu, y}\big[\big(\mathbf{x}^\top \mathbf{w} - y(\mathbf{x})\big)^2\big] + \lambda \|\mathbf{w}\|^2 \right\} = (\boldsymbol{\Sigma}_\mu + \lambda \mathbf{I})^{-1} \mathbf{v}_{\mu, y}, \quad \text{where } \mathbf{v}_{\mu, y} = \mathbb{E}_{\mu, y}[y(\mathbf{x})\,\mathbf{x}].
$$

When Assumption 1 holds, then $\mathbf{v}_{\mu, y} = \boldsymbol{\Sigma}_\mu \mathbf{w}^*$, however ridge-regularized least squares is well-defined for much more general response models. Our second result makes a direct connection between the (expectation of the) unregularized minimum norm solution on the sample and the global ridge-regularized solution. While the under-determined regime (i.e., $n < d$) is of primary interest to us, for completeness we state this result for arbitrary values of $n$ and $d$. Note that, just like the definition of regularized least squares, this theorem applies more generally than Theorem 1, in that it does *not* require the responses to follow any linear model as in Assumption 1 (proof in Appendix D).

**Theorem 2 (Implicit regularization of Moore-Penrose estimator)** *For $\mu$ satisfying Assumption 2 and $y(\cdot)$ s.t. $\mathbf{v}_{\mu,y} = \mathbb{E}_{\mu,y}[y(\mathbf{x})\,\mathbf{x}]$ is well-defined, $\bar{\mathbf{X}} \sim S_\mu^n$ (Definition 3) and $\bar{y}_i = y(\bar{\mathbf{x}}_i)$,*

$$\mathbb{E}\big[\bar{\mathbf{X}}^\dagger \bar{\mathbf{y}}\big] = \begin{cases} (\mathbf{\Sigma}_\mu + \lambda_n \mathbf{I})^{-1} \mathbf{v}_{\mu,y} & \text{for } n < d, \\ \mathbf{\Sigma}_\mu^{-1} \mathbf{v}_{\mu,y} & \text{for } n \geq d, \end{cases}$$

*where, as in Theorem 1, $\lambda_n$ is such that the effective dimension $\mathrm{tr}(\mathbf{\Sigma}_\mu(\mathbf{\Sigma}_\mu + \lambda_n \mathbf{I})^{-1})$ equals $n$.*

That is, when $n < d$, the Moore-Penrose estimator (which itself is not regularized), computed on the random training sample, in expectation equals the global ridge-regularized least squares solution of the underlying regression problem. Moreover, $\lambda_n$, i.e., the amount of implicit $\ell_2$-regularization, is controlled by the degree of over-parameterization in such a way as to ensure that $n$ becomes the ridge effective dimension (a.k.a. the effective degrees of freedom).

We illustrate this result in Figure 1b, plotting the norm of the expectation of the Moore-Penrose estimator. As for the MSE, our surrogate theory aligns well with the empirical estimates for i.i.d. Gaussian designs, showing that the shrinkage of the unregularized estimator in the under-determined regime matches the implicit ridge-regularization characterized by Theorem 2. While the shrinkage is a linear function of the sample size $n$ for isotropic features (i.e., $\mathbf{\Sigma} = \mathbf{I}$), it exhibits a non-linear behavior for other spectral decays. Such *implicit regularization* has been studied previously [see, e.g., MO11, Mah12]; it has been observed empirically for RandNLA sampling algorithms [MMY15]; and it has also received attention more generally within the context of neural networks [Ney17]. While our implicit regularization result is limited to the Moore-Penrose estimator, this new connection (and others, described below) between the minimum norm solution of an unregularized under-determined system and a ridge-regularized least squares solution offers a simple interpretation for the implicit regularization observed in modern machine learning architectures.

Our exact non-asymptotic expressions in Theorem 1 and our exact implicit regularization results in Theorem 2 are derived for the surrogate design, which is a non-i.i.d. distribution based on a determinantal point process. However, Figure 1 suggests that those expressions accurately describe the MSE (up to lower order terms) also under the standard i.i.d. design $\mathbf{X} \sim \mu^n$ when $\mu$ is a multivariate Gaussian. As a third result, we verify that the surrogate expressions for the MSE are asymptotically consistent with the MSE of an i.i.d. design, for a wide class of distributions which include multivariate Gaussians.

**Theorem 3 (Asymptotic consistency of surrogate design)** *Let $\mathbf{X} \in \mathbb{R}^{n \times d}$ have i.i.d. rows $\mathbf{x}_i^\top = \mathbf{z}_i^\top \mathbf{\Sigma}^{\frac{1}{2}}$ where $\mathbf{z}_i$ has independent zero mean and unit variance sub-Gaussian entries, and suppose that Assumptions 1 and 2 are satisfied. Furthermore, suppose that there exist $c, C, C^* \in \mathbb{R}_{>0}$ such that $C\mathbf{I} \succeq \mathbf{\Sigma} \succeq c\mathbf{I} \succ 0$ and $\|\mathbf{w}^*\| \leq C^*$. Then*

$$\mathrm{MSE}\big[\mathbf{X}^\dagger \mathbf{y}\big] - \mathcal{M}(\mathbf{\Sigma}, \mathbf{w}^*, \sigma^2, n) \to 0$$

*with probability one as $d, n \to \infty$ with $n/d \to \bar{c} \in (0, \infty) \setminus \{1\}$.*

The above result is particularly remarkable since our surrogate design is a determinantal point process. DPPs are commonly used in ML to ensure that the data points in a sample are well spread-out. However, if the data distribution is sufficiently regular (e.g., a multivariate Gaussian), then the i.i.d. samples are already spread-out reasonably well, so rescaling the distribution by a determinant has a negligible effect that vanishes in the high-dimensional regime. Furthermore, our empirical estimates (Figure 1) suggest that the surrogate expressions are accurate not only in the asymptotic limit, but even for moderately large dimensions. Based on a detailed empirical analysis described in Section 5, we conjecture that the convergence described in Theorem 3 has the rate of $O(1/d)$.

## 2 Related work

There is a large body of related work, which for simplicity we cluster into three groups.

**Double descent.** The double descent phenomenon has been observed empirically in a number of learning models, including neural networks [BHMM19, GJS$^+$19], kernel methods [BMM18, BRT19], nearest neighbor models [BHM18], and decision trees [BHMM19]. The theoretical analysis of double descent, and more broadly the generalization properties of interpolating estimators, have primarily

focused on various forms of linear regression [BLLT19, LR19, HMRT19, MVSS19]. Note that while we analyze the classical mean squared error, many works focus on the squared prediction error. Also, unlike in our work, some of the literature on double descent deals with linear regression in the so-called *misspecified* setting, where the set of observed features does not match the feature space in which the response model is linear [BHX19, HMRT19, Mit19, MM19b], e.g., when the learner observes a random subset of $d$ features from a larger population.

The most directly comparable to our setting is the recent work of [HMRT19]. They study how varying the feature dimension affects the (asymptotic) generalization error for linear regression, however their analysis is limited to certain special settings such as an isotropic data distribution. As an additional point of comparison, in Figure 2 we plot the MSE expressions of Theorem 1 when varying the feature dimension $d$ (the setup is the same as in Figure 1). Our plots follow the trends outlined by [HMRT19] for the isotropic case (see their Figure 2), but the spectral decay of the covariance (captured by our new MSE expressions) has a significant effect on the descent curve. This leads to generalization in the under-determined regime even when the signal-to-noise ratio (SNR = $\|\mathbf{w}^*\|^2/\sigma^2$) is 1, unlike suggested by [HMRT19].

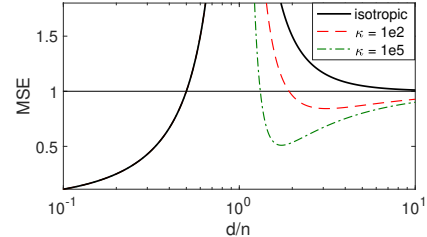

Figure 2: Surrogate MSE as a function of $d/n$, with $n$ fixed to 100 and varying dimension $d$ and condition number $\kappa$, for signal-to-noise ratio SNR = 1.

**RandNLA and DPPs.** Randomized Numerical Linear Algebra [DM16, DM17] has traditionally focused on obtaining purely algorithmic improvements for tasks such as least squares regression, but there has been growing interest in understanding the statistical properties of these randomized methods [MMY15, RM16] and a beyond worst-case analysis [DLLM20]. Determinantal point processes [DM20, KT12] have been recently shown to combine strong worst-case regression guarantees with elegant statistical properties [DW17]. However, these results are limited to the over-determined setting [DWH18, DWH19a, DCMW19] and ridge regression [DW18, DLM20]. Our results are also related to recent work on using DPPs to analyze the expectation of the inverse [DM19, DBPM20] and generalized inverse [MDK20, DKM20] of a subsampled matrix.

**Implicit regularization.** The term implicit regularization typically refers to the notion that approximate computation can implicitly lead to statistical regularization. See [MO11, PM11, GM14] and references therein for early work on the topic; and see [Mah12] for an overview. More recently, often motivated by neural networks, there has been work on implicit regularization that typically considered SGD-based optimization algorithms. See, e.g., theoretical results [NTS14, Ney17, SHN+18, GWB+17, ACHL19, KBMM19] as well as extensive empirical studies [MM18, MM19a]. The implicit regularization observed by us is different in that it is not caused by an inexact approximation algorithm (such as SGD) but rather by the selection of one out of many exact solutions (e.g., the minimum norm solution). In this context, most relevant are the asymptotic results of [KLS18] and [LJB19].

## 3  Surrogate random designs

In this section, we provide the definition of our surrogate random design $S_\mu^n$, where $\mu$ is a $d$-variate probability measure and $n$ is the sample size. This distribution is used in place of the standard random design $\mu^n$ consisting of $n$ row vectors drawn independently from $\mu$.

**Preliminaries.** For an $n \times n$ matrix $\mathbf{A}$, we use $\mathrm{pdet}(\mathbf{A})$ to denote the pseudo-determinant of $\mathbf{A}$, which is the product of non-zero eigenvalues (repeated eigenvalues are taken to the power of their algebraic multiplicity). For index subsets $\mathcal{I}$ and $\mathcal{J}$, we use $\mathbf{A}_{\mathcal{I},\mathcal{J}}$ to denote the submatrix of $\mathbf{A}$ with rows indexed by $\mathcal{I}$ and columns indexed by $\mathcal{J}$. We may write $\mathbf{A}_{\mathcal{I},*}$ to indicate that we take a subset of rows. We let $\mathbf{X} \sim \mu^k$ denote a $k \times d$ random matrix with rows drawn i.i.d. according to $\mu$, and the $i$th row is denoted as $\mathbf{x}_i^\top$. We also let $\boldsymbol{\Sigma}_\mu = \mathbb{E}_\mu[\mathbf{x}\mathbf{x}^\top]$, where $\mathbb{E}_\mu$ refers to the expectation with respect to $\mathbf{x}^\top \sim \mu$, assuming throughout that $\boldsymbol{\Sigma}_\mu$ is well-defined and positive definite. We use $\mathrm{Poisson}(\gamma)_{\leq a}$ as the Poisson distribution restricted to $[0, a]$, whereas $\mathrm{Poisson}(\gamma)_{\geq a}$ is restricted to $[a, \infty)$. We also let $\#(\mathbf{X})$ denote the number of rows of $\mathbf{X}$.

**Definition 2** *Let $\mu$ satisfy Assumption 2 and let $K$ be a random variable over $\mathbb{Z}_{\geq 0}$. A determinantal design $\bar{\mathbf{X}} \sim \mathrm{Det}(\mu, K)$ is a distribution with the same domain as $\mathbf{X} \sim \mu^K$ such that for any event $E$ measurable w.r.t. $\mathbf{X}$, we have*

$$\mathrm{Pr}\{\bar{\mathbf{X}} \in E\} = \frac{\mathbb{E}[\mathrm{pdet}(\mathbf{X}\mathbf{X}^\top)\mathbf{1}_{[\mathbf{X}\in E]}]}{\mathbb{E}[\mathrm{pdet}(\mathbf{X}\mathbf{X}^\top)]}.$$

The above definition can be interpreted as rescaling the density function of $\mu^K$ by the pseudo-determinant, and then renormalizing it. We now construct our surrogate design $S_\mu^n$ by appropriately selecting the random variable $K$. The obvious choice of $K = n$ does *not* result in simple closed form expressions for the MSE in the under-determined regime (i.e., $n < d$), which is the regime of primary interest to us. Instead, we derive our random variables $K$ from the Poisson distribution.

**Definition 3** *For $\mu$ satisfying Assumption 2, define surrogate design $S_\mu^n$ as $\mathrm{Det}(\mu, K)$ where:*

1. *if $n < d$, then $K \sim \mathrm{Poisson}(\gamma_n)_{\leq d}$ with $\gamma_n$ as the solution of $n = \mathrm{tr}(\mathbf{\Sigma}_\mu(\mathbf{\Sigma}_\mu + \frac{1}{\gamma_n}\mathbf{I})^{-1})$,*
2. *if $n = d$, then we simply let $K = d$,*
3. *if $n > d$, then $K \sim \mathrm{Poisson}(\gamma_n)_{\geq d}$ with $\gamma_n = n - d$.*

Note that the under-determined case, i.e., $n < d$, is restricted to $K \leq d$ so that, under Assumption 2, $\mathrm{pdet}(\mathbf{X}\mathbf{X}^\top) = \det(\mathbf{X}\mathbf{X}^\top)$ with probability 1. On the other hand in the over-determined case, i.e., $n > d$, we have $K \geq d$ so that $\mathrm{pdet}(\mathbf{X}\mathbf{X}^\top) = \det(\mathbf{X}^\top\mathbf{X})$. In the special case of $n = d = K$ both of these equations are satisfied: $\mathrm{pdet}(\mathbf{X}\mathbf{X}^\top) = \det(\mathbf{X}^\top\mathbf{X}) = \det(\mathbf{X}\mathbf{X}^\top) = \det(\mathbf{X})^2$.

The first non-trivial property of the surrogate design $S_\mu^n$ is that the expected sample size is in fact always equal to $n$, which we prove in Appendix A.

**Lemma 1** *Let $\bar{\mathbf{X}} \sim S_\mu^n$ for any $n > 0$. Then, we have $\mathbb{E}[\#(\bar{\mathbf{X}})] = n$.*

Our general template for computing expectations under a surrogate design $\bar{\mathbf{X}} \sim \mathbf{S}_\mu^n$ is to use the following expressions based on the i.i.d. random design $\mathbf{X} \sim \mu^K$:

$$\mathbb{E}[F(\bar{\mathbf{X}})] = \begin{cases} \frac{\mathbb{E}[\det(\mathbf{X}\mathbf{X}^\top)F(\mathbf{X})]}{\mathbb{E}[\det(\mathbf{X}\mathbf{X}^\top)]} & K \sim \mathrm{Poisson}(\gamma_n) & \text{for } n < d, \\ \frac{\mathbb{E}[\det(\mathbf{X})^2 F(\mathbf{X})]}{\mathbb{E}[\det(\mathbf{X})^2]} & K = d & \text{for } n = d, \\ \frac{\mathbb{E}[\det(\mathbf{X}^\top\mathbf{X})F(\mathbf{X})]}{\mathbb{E}[\det(\mathbf{X}^\top\mathbf{X})]} & K \sim \mathrm{Poisson}(\gamma_n) & \text{for } n > d. \end{cases} \tag{1}$$

These formulas follow from Definitions 2 and 3 because the determinants $\det(\mathbf{X}\mathbf{X}^\top)$ and $\det(\mathbf{X}^\top\mathbf{X})$ are non-zero precisely in the regimes $n \leq d$ and $n \geq d$, respectively, which is why we can drop the restrictions on the range of the Poisson distribution. We compute the normalization constants by introducing the concept of determinant preserving random matrices, discussed in Section 4.

**Proof sketch of Theorem 1** We focus here on the under-determined regime (i.e., $n < d$), high-lighting the key new expectation formulas we develop to derive the MSE expressions for surrogate designs. A standard decomposition of the MSE yields:

$$\mathrm{MSE}[\bar{\mathbf{X}}^\dagger\bar{\mathbf{y}}] = \mathbb{E}[\|\bar{\mathbf{X}}^\dagger(\bar{\mathbf{X}}\mathbf{w}^* + \boldsymbol{\xi}) - \mathbf{w}^*\|^2] = \sigma^2\mathbb{E}[\mathrm{tr}((\bar{\mathbf{X}}^\top\bar{\mathbf{X}})^\dagger)] + \mathbf{w}^{*\top}\mathbb{E}[\mathbf{I} - \bar{\mathbf{X}}^\dagger\bar{\mathbf{X}}]\mathbf{w}^*. \tag{2}$$

Thus, our task is to find closed form expressions for the two expectations above. The latter, which is the expected projection onto the complement of the row-span of $\bar{\mathbf{X}}$, is proven in Appendix D.

**Lemma 2** *If $\bar{\mathbf{X}} \sim S_\mu^n$ and $n < d$, then we have: $\mathbb{E}[\mathbf{I} - \bar{\mathbf{X}}^\dagger\bar{\mathbf{X}}] = (\gamma_n\mathbf{\Sigma}_\mu + \mathbf{I})^{-1}$.*

No such expectation formula is known for i.i.d. designs, except when $\mu$ is an isotropic Gaussian. In Appendix D, we also prove a generalization of Lemma 2 which is then used to establish our implicit regularization result (Theorem 2). We next give an expectation formula for the trace of the Moore-Penrose inverse of the covariance matrix for a surrogate design (proof in Appendix C).

**Lemma 3** *If $\bar{\mathbf{X}} \sim S_\mu^n$ and $n < d$, then: $\mathbb{E}[\mathrm{tr}((\bar{\mathbf{X}}^\top\bar{\mathbf{X}})^\dagger)] = \gamma_n\big(1 - \det\big((\frac{1}{\gamma_n}\mathbf{I} + \mathbf{\Sigma}_\mu)^{-1}\mathbf{\Sigma}_\mu\big)\big)$.*

Note the implicit regularization term which appears in both formulas, given by $\lambda_n = \frac{1}{\gamma_n}$. Since $n = \mathrm{tr}(\mathbf{\Sigma}_\mu(\mathbf{\Sigma}_\mu + \lambda_n\mathbf{I})^{-1}) = d - \lambda_n\mathrm{tr}((\mathbf{\Sigma}_\mu + \lambda_n\mathbf{I})^{-1})$, it follows that $\lambda_n = (d-n)/\mathrm{tr}((\mathbf{\Sigma}_\mu + \lambda_n\mathbf{I})^{-1})$. Combining this with Lemmas 2 and 3, we recover the surrogate MSE expression in Theorem 1.

# 4 Determinant preserving random matrices

In this section, we introduce the key tool for computing expectation formulas of matrix determinants. It is used in our analysis of the surrogate design, and it should be of independent interest.

The key question motivating the following definition is: *When does taking expectation commute with computing a determinant for a square random matrix?*

**Definition 4** *A random $d \times d$ matrix $\mathbf{A}$ is called determinant preserving (d.p.), if*

$$\mathbb{E}\big[\det(\mathbf{A}_{\mathcal{I},\mathcal{J}})\big] = \det\big(\mathbb{E}[\mathbf{A}_{\mathcal{I},\mathcal{J}}]\big) \quad \textit{for all } \mathcal{I}, \mathcal{J} \subseteq [d] \textit{ s.t. } |\mathcal{I}| = |\mathcal{J}|.$$

We next give a few simple examples to provide some intuition. First, note that every $1 \times 1$ random matrix is determinant preserving simply because taking a determinant is an identity transfomation in one dimension. Similarly, every fixed matrix is determinant preserving because in this case taking the expectation is an identity transformation. In all other cases, however, Definition 4 has to be verified more carefully. Further examples (positive and negative) follow.

**Example 1** *If $\mathbf{A}$ has i.i.d. Gaussian entries $a_{ij} \sim \mathcal{N}(0, 1)$, then $\mathbf{A}$ is d.p. because $\mathbb{E}[\det(\mathbf{A})] = 0$.*

In fact, it can be shown that all random matrices with independent entries are determinant preserving. However, this is not a necessary condition.

**Example 2** *Let $\mathbf{A} = s\,\mathbf{Z}$, where $\mathbf{Z}$ is fixed with $\mathrm{rank}(\mathbf{Z}) = r$, and $s$ is a scalar random variable. Then for $|\mathcal{I}| = |\mathcal{J}| = r$ we have*

$$\mathbb{E}\big[\det(s\,\mathbf{Z}_{\mathcal{I},\mathcal{J}})\big] = \mathbb{E}[s^r]\det(\mathbf{Z}_{\mathcal{I},\mathcal{J}}) = \det\left(\big(\mathbb{E}[s^r]\big)^{\frac{1}{r}} \mathbf{Z}_{\mathcal{I},\mathcal{J}}\right),$$

*so if $r = 1$ then $\mathbf{A}$ is determinant preserving, whereas if $r > 1$ and $\mathrm{Var}[s] > 0$ then it is not.*

To construct more complex examples, we show that determinant preserving random matrices are closed under addition and multiplication. The proof of this result is an extension of an existing argument, given by [DM19] in the proof of Lemma 7, for computing the expected determinant of the sum of rank-1 random matrices (proof in Appendix B).

**Lemma 4 (Closure properties)** *If $\mathbf{A}$ and $\mathbf{B}$ are independent and determinant preserving, then:*

1. *$\mathbf{A} + \mathbf{B}$ is determinant preserving,*
2. *$\mathbf{AB}$ is determinant preserving.*

Next, we introduce another important class of d.p. matrices: a sum of i.i.d. rank-1 random matrices with the number of i.i.d. samples being a Poisson random variable. Our use of the Poisson distribution is crucial for the below result to hold. It is an extension of an expectation formula given by [Der19] for sampling from discrete distributions (proof in Appendix B).

**Lemma 5** *If $K$ is a Poisson random variable and $\mathbf{A}, \mathbf{B}$ are random $K \times d$ matrices whose rows are sampled as an i.i.d. sequence of joint pairs of random vectors, then $\mathbf{A}^\top \mathbf{B}$ is d.p., and so:*

$$\mathbb{E}\big[\det(\mathbf{A}^\top \mathbf{B})\big] = \det\big(\mathbb{E}[\mathbf{A}^\top \mathbf{B}]\big).$$

Finally, we show the expectation formula needed for obtaining the normalization constant of the under-determined surrogate design, given in (1). The below result is more general than the normalization constant requires, because it allows the matrices $\mathbf{A}$ and $\mathbf{B}$ to be different (the constant is obtained by setting $\mathbf{A} = \mathbf{B} = \mathbf{X} \sim \mu^K$). In fact, we use this more general statement to show Theorems 1 and 2. The proof uses Lemmas 4 and 5 (see Appendix B).

**Lemma 6** *If $K$ is a Poisson random variable and $\mathbf{A}$, $\mathbf{B}$ are random $K \times d$ matrices whose rows are sampled as an i.i.d. sequence of joint pairs of random vectors, then*

$$\mathbb{E}\big[\det(\mathbf{AB}^\top)\big] = \mathrm{e}^{-\mathbb{E}[K]}\det\big(\mathbf{I} + \mathbb{E}[\mathbf{B}^\top \mathbf{A}]\big).$$

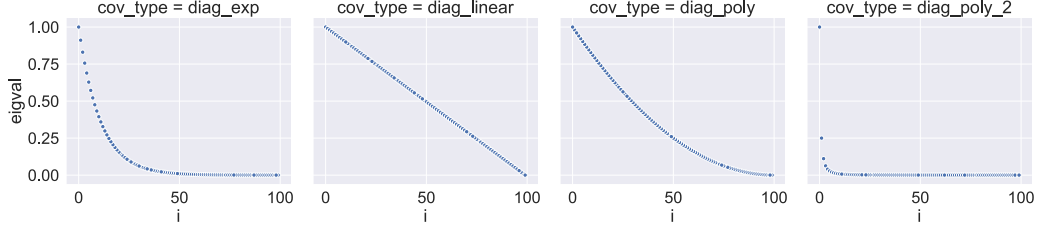

Figure 3: Scree-plots of $\boldsymbol{\Sigma}$ for the eigenvalue decays examined in our empirical valuations.

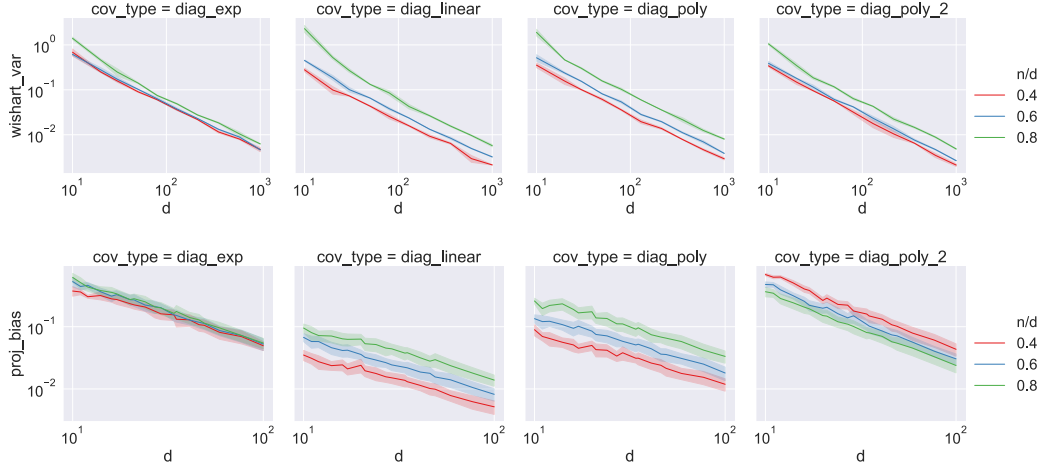

Figure 4: Empirical verification of the asymptotic consistency of surrogate MSE. We show the discrepancies for the variance (top) and bias (bottom), with bootstrapped 95% confidence intervals, as $d$ increases and $n/d$ is fixed. We observe $O(1/d)$ decay (linear with slope $-1$ on a log-log plot).

## 5  Empirical evaluation of asymptotic consistency

In this section, we empirically quantify the convergence rates for the asymptotic result of Theorem 3. We focus on the under-determined regime (i.e., $n < d$ and separate the evaluation into the bias and variance terms, following the MSE decomposition given in (2). Consider $\mathbf{X} = \mathbf{Z}\boldsymbol{\Sigma}^{1/2}$, where the entries of $\mathbf{Z}$ are i.i.d. standard Gaussian, and define:

1. Variance discrepancy: $\left|\frac{\mathbb{E}[\mathrm{tr}((\mathbf{X}^\top\mathbf{X})^\dagger)]}{\mathcal{V}(\boldsymbol{\Sigma},n)} - 1\right|$ where $\mathcal{V}(\boldsymbol{\Sigma},n) = \frac{1-\alpha_n}{\lambda_n}$.

2. Bias discrepancy: $\sup_{\mathbf{w}\in\mathbb{R}^d\setminus\{\mathbf{0}\}} \left|\frac{\mathbf{w}^\top\mathbb{E}[\mathbf{I}-\mathbf{X}^\dagger\mathbf{X}]\mathbf{w}}{\mathbf{w}^\top\mathcal{B}(\boldsymbol{\Sigma},n)\mathbf{w}} - 1\right|$ where $\mathcal{B}(\boldsymbol{\Sigma},n) = \lambda_n(\boldsymbol{\Sigma}+\lambda_n\mathbf{I})^{-1}$.

Recall that $\lambda_n = \frac{d-n}{\mathrm{tr}((\boldsymbol{\Sigma}+\lambda_n\mathbf{I})^{-1})}$, so our surrogate MSE can be written as $\mathcal{M} = \sigma^2\mathcal{V}(\boldsymbol{\Sigma},n) + \mathbf{w}^{*\top}\mathcal{B}(\boldsymbol{\Sigma},n)\mathbf{w}^*$, and when both discrepancies are bounded by $\epsilon$, then $(1-2\epsilon)\mathcal{M} \leq \mathrm{MSE}\big[\mathbf{X}^\dagger\mathbf{y}\big] \leq (1+2\epsilon)\mathcal{M}$. In our experiments, we consider four standard eigenvalue decay profiles for $\boldsymbol{\Sigma}$, including polynomial and exponential decay (see Figure 3 and Appendix F.1).

Figure 4 (top) plots the variance discrepancy (with $\mathbb{E}[\mathrm{tr}((\mathbf{X}^\top\mathbf{X})^\dagger)]$ estimated via Monte Carlo sampling and bootstrapped confidence intervals) as $d$ increases from 10 to 1000, across a range of aspect ratios $n/d$. In all cases, we observe that the discrepancy decays to zero at a rate of $O(1/d)$. Figure 4 (bottom) plots the bias discrepancy, with the same rate of decay observed throughout. Note that the range of $d$ is smaller than in Figure 4 (top) because the large number of Monte Carlo samples (up to two million) required for this experiment made the computations much more expensive (more details in Appendix F). Based on the above empirical results, we conclude with a conjecture.

**Conjecture 1** *When $\mu$ is a centered multivariate Gaussian and its covariance has a constant condition number, then, for $n/d$ fixed, the surrogate MSE satisfies:* $\left|\frac{\mathrm{MSE}[\mathbf{X}^\dagger\mathbf{y}]}{\mathcal{M}} - 1\right| = O(1/d)$.

# 6 Conclusions

We derived exact non-asymptotic expressions for the MSE of the Moore-Penrose estimator in the linear regression task, reproducing the double descent phenomenon as the sample size crosses between the under- and over-determined regime. To achieve this, we modified the standard i.i.d. random design distribution using a determinantal point process to obtain a surrogate design which admits exact MSE expressions, while capturing the key properties of the i.i.d. design. We also provided a result that relates the expected value of the Moore-Penrose estimator of a training sample in the under-determined regime (i.e., the minimum norm solution) to the ridge-regularized least squares solution for the population distribution, thereby providing an interpretation for the implicit regularization resulting from over-parameterization.

## Broader Impact

While the double descent phenomenon has been empirically observed in a variety of applications, mathematical descriptions of it are oftentimes complex and inaccesible to non-experts. In contrast, our surrogate design and the accompanying closed-form expressions provide an easily computable rule of thumb for estimating the generalization error. One important application is the high-dimensional (i.e. underdetermined, $n < d$) regime experienced by modern machine learning systems where the number of parameters vastly exceeds the quantity of available data. Our research can be applied here to provide a theoretical understanding of the surprising phenomenon where without proper regularization [NVKM20] more data (i.e. increasing $n$) may lead to worse generalization performance.

Better theoretical understanding of generalization error in the small data regime has important societal impact. Through theoretically modeling a system's performance, we can build safer systems by better understanding how badly an estimator fails and accounting for these failure modes in the system's design. Furthermore, the improved understanding of minimum norm solutions performing worse with more data offers an appealing trade-off where certain systems can both improve their performance and respect the privacy of its users by collecting less data.

**Acknowledgements.** We would like to acknowledge ARO, DARPA, NSF, ONR, and GFSD for providing partial support of this work. We also thank Zhenyu Liao for pointing out fruitful connections between our results and the asymptotic analysis of random matrix resolvents.

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
