[Supplementary Material]

# A  Proof of Lemma 1

We first record an important property of the design $S_\mu^d$ which can be used to construct an over-determined design for any $n > d$. A similar version of this result was also previously shown by [DWH19b] for a different determinantal design.

**Lemma 7** *Let $\bar{\mathbf{X}} \sim S_\mu^d$ and $\mathbf{X} \sim \mu^K$, where $K \sim \mathrm{Poisson}(\gamma)$. Then the matrix composed of a random permutation of the rows from $\bar{\mathbf{X}}$ and $\mathbf{X}$ is distributed according to $S_\mu^{d+\gamma}$.*

**Proof** Let $\widetilde{\mathbf{X}}$ denote the matrix constructed from the permuted rows of $\bar{\mathbf{X}}$ and $\mathbf{X}$. Letting $\mathbf{Z} \sim \mu^{K+d}$, we derive the probability $\Pr\{\widetilde{\mathbf{X}} \in E\}$ by summing over the possible index subsets $S \subseteq [K + d]$ that correspond to the rows coming from $\bar{\mathbf{X}}$:

$$
\begin{aligned}
\Pr\{\widetilde{\mathbf{X}} \in E\} &= \mathbb{E}\left[ \frac{1}{\binom{K+d}{d}} \sum_{S:\, |S|=d} \frac{\mathbb{E}[\det(\mathbf{Z}_{S,*})^2 \mathbf{1}_{[\mathbf{Z} \in E]} \mid K]}{d!\det(\mathbf{\Sigma}_\mu)} \right] \\
&= \sum_{k=0}^{\infty} \frac{\gamma^k \mathrm{e}^{-\gamma}}{k!} \frac{\gamma^d k!}{(k+d)!} \frac{\mathbb{E}\big[ \sum_{S:\, |S|=d} \det(\mathbf{Z}_{S,*})^2 \mathbf{1}_{[\mathbf{Z} \in E]} \mid K = k \big]}{\det(\gamma\mathbf{\Sigma}_\mu)} \\
&\stackrel{(*)}{=} \sum_{k=0}^{\infty} \frac{\gamma^{k+d} \mathrm{e}^{-\gamma}}{(k+d)!} \frac{\mathbb{E}[\det(\mathbf{Z}^\top \mathbf{Z}) \mathbf{1}_{[\mathbf{Z} \in E]} \mid K = k]}{\det(\gamma\mathbf{\Sigma}_\mu)},
\end{aligned}
$$

where $(*)$ uses the Cauchy-Binet formula to sum over all subsets $S$ of size $d$. Finally, since the sum shifts from $k$ to $k + d$, the last expression can be rewritten as $\mathbb{E}[\det(\mathbf{X}^\top \mathbf{X}) \mathbf{1}_{[\mathbf{X} \in E]}]/\det(\gamma\mathbf{\Sigma}_\mu)$, where recall that $\mathbf{X} \sim \mu^K$ and $K \sim \mathrm{Poisson}(\gamma)$, matching the definition of $S_\mu^{d+\gamma}$. $\blacksquare$

We now proceed with the proof of Lemma 1, where we establish that the expected sample size of $S_\mu^n$ is indeed $n$.

**Proof of Lemma 1** The result is obvious when $n = d$, whereas for $n > d$ it is an immediate consequence of Lemma 7. Finally, for $n < d$ the expected sample size follows as a corollary of Lemma 2, which states that

$$
\text{(Lemma 2)} \qquad \mathbb{E}\big[\mathbf{I} - \bar{\mathbf{X}}^\dagger \bar{\mathbf{X}}\big] = (\gamma_n \mathbf{\Sigma}_\mu + \mathbf{I})^{-1},
$$

where $\bar{\mathbf{X}}^\dagger \bar{\mathbf{X}}$ is the orthogonal projection onto the subspace spanned by the rows of $\bar{\mathbf{X}}$. Since the rank of this subspace is equal to the number of the rows, we have $\#(\bar{\mathbf{X}}) = \mathrm{tr}(\bar{\mathbf{X}}^\dagger \bar{\mathbf{X}})$, so

$$
\mathbb{E}\big[\#(\bar{\mathbf{X}})\big] = d - \mathrm{tr}\big((\gamma_n \mathbf{\Sigma}_\mu + \mathbf{I})^{-1}\big) = \mathrm{tr}\big(\gamma_n \mathbf{\Sigma}_\mu(\gamma_n \mathbf{\Sigma}_\mu + \mathbf{I})^{-1}\big) = n,
$$

which completes the proof. $\blacksquare$

# B  Proofs for Section 4

We use $\mathrm{adj}(\mathbf{A})$ to denote the adjugate of $\mathbf{A}$, defined as follows: the $(i, j)$th entry of $\mathrm{adj}(\mathbf{A})$ is $(-1)^{i+j} \det(\mathbf{A}_{[n]\setminus\{j\}, [n]\setminus\{i\}})$. We will use two useful identities related to the adjugate: (1) $\mathrm{adj}(\mathbf{A}) = \det(\mathbf{A})\mathbf{A}^{-1}$ for invertible $\mathbf{A}$, and (2) $\det(\mathbf{A} + \mathbf{u}\mathbf{v}^\top) = \det(\mathbf{A}) + \mathbf{v}^\top \mathrm{adj}(\mathbf{A})\mathbf{u}$ [see Fact 2.14.2 in Ber11].

First, note that from the definition of an adjugate matrix it immediately follows that if $\mathbf{A}$ is determinant preserving then adjugate commutes with expectation for this matrix:

$$
\begin{aligned}
\mathbb{E}\big[\big(\mathrm{adj}(\mathbf{A})\big)_{i,j}\big] &= \mathbb{E}\big[(-1)^{i+j} \det(\mathbf{A}_{[d]\setminus\{j\}, [d]\setminus\{i\}})\big] \\
&= (-1)^{i+j} \det\big(\mathbb{E}[\mathbf{A}_{[d]\setminus\{j\}, [d]\setminus\{i\}}]\big) \qquad (3) \\
&= \big(\mathrm{adj}(\mathbb{E}[\mathbf{A}])\big)_{i,j}. \qquad (4)
\end{aligned}
$$

**Proof of Lemma 4**  First, we show that $\mathbf{A} + \mathbf{u}\mathbf{v}^\top$ is d.p. for any fixed $\mathbf{u}, \mathbf{v} \in \mathbb{R}^d$. Below, we use the identity for a rank one update of a determinant: $\det(\mathbf{A} + \mathbf{u}\mathbf{v}^\top) = \det(\mathbf{A}) + \mathbf{v}^\top \mathrm{adj}(\mathbf{A})\mathbf{u}$. It follows that for any $\mathcal{I}$ and $\mathcal{J}$ of the same size,

$$
\begin{aligned}
\mathbb{E}\big[\det(\mathbf{A}_{\mathcal{I},\mathcal{J}} + \mathbf{u}_{\mathcal{I}}\mathbf{v}_{\mathcal{J}}^\top)\big] &= \mathbb{E}\big[\det(\mathbf{A}_{\mathcal{I},\mathcal{J}}) + \mathbf{v}_{\mathcal{J}}^\top \mathrm{adj}(\mathbf{A}_{\mathcal{I},\mathcal{J}})\mathbf{u}_{\mathcal{I}}\big] \\
&\overset{(*)}{=} \det\big(\mathbb{E}[\mathbf{A}_{\mathcal{I},\mathcal{J}}]\big) + \mathbf{v}_{\mathcal{J}}^\top \mathrm{adj}\big(\mathbb{E}[\mathbf{A}_{\mathcal{I},\mathcal{J}}]\big)\mathbf{u}_{\mathcal{I}} \\
&= \det\big(\mathbb{E}[\mathbf{A}_{\mathcal{I},\mathcal{J}} + \mathbf{u}_{\mathcal{I}}\mathbf{v}_{\mathcal{J}}^\top]\big),
\end{aligned}
$$

where $(*)$ used (4), i.e., the fact that for d.p. matrices, adjugate commutes with expectation. Crucially, through the definition of an adjugate this step implicitly relies on the assumption that all the square submatrices of $\mathbf{A}_{\mathcal{I},\mathcal{J}}$ are also determinant preserving. Iterating this, we get that $\mathbf{A} + \mathbf{Z}$ is d.p. for any fixed $\mathbf{Z}$. We now show the same for $\mathbf{A} + \mathbf{B}$:

$$
\begin{aligned}
\mathbb{E}\big[\det(\mathbf{A}_{\mathcal{I},\mathcal{J}} + \mathbf{B}_{\mathcal{I},\mathcal{J}})\big] &= \mathbb{E}\Big[\mathbb{E}\big[\det(\mathbf{A}_{\mathcal{I},\mathcal{J}} + \mathbf{B}_{\mathcal{I},\mathcal{J}}) \mid \mathbf{B}\big]\Big] \\
&\overset{(*)}{=} \mathbb{E}\Big[\det\big(\mathbb{E}[\mathbf{A}_{\mathcal{I},\mathcal{J}}] + \mathbf{B}_{\mathcal{I},\mathcal{J}}\big)\Big] \\
&= \det\big(\mathbb{E}[\mathbf{A}_{\mathcal{I},\mathcal{J}} + \mathbf{B}_{\mathcal{I},\mathcal{J}}]\big),
\end{aligned}
$$

where $(*)$ uses the fact that after conditioning on $\mathbf{B}$ we can treat it as a fixed matrix. Next, we show that $\mathbf{A}\mathbf{B}$ is determinant preserving via the Cauchy-Binet formula:

$$
\begin{aligned}
\mathbb{E}\big[\det\big((\mathbf{A}\mathbf{B})_{\mathcal{I},\mathcal{J}}\big)\big] &= \mathbb{E}\big[\det(\mathbf{A}_{\mathcal{I},*}\mathbf{B}_{*,\mathcal{J}})\big] \\
&= \mathbb{E}\Bigg[\sum_{S:\,|S|=|\mathcal{I}|} \det\big(\mathbf{A}_{\mathcal{I},S}\big)\det\big(\mathbf{B}_{S,\mathcal{J}}\big)\Bigg] \\
&= \sum_{S:\,|S|=|\mathcal{I}|} \det\big(\mathbb{E}[\mathbf{A}]_{\mathcal{I},S}\big)\det\big(\mathbb{E}[\mathbf{B}]_{S,\mathcal{J}}\big) \\
&= \det\big(\mathbb{E}[\mathbf{A}]_{\mathcal{I},*}\,\mathbb{E}[\mathbf{B}]_{*,\mathcal{J}}\big) \\
&= \det\big(\mathbb{E}[\mathbf{A}\mathbf{B}]_{\mathcal{I},\mathcal{J}}\big),
\end{aligned}
$$

where recall that $\mathbf{A}_{\mathcal{I},*}$ denotes the submatrix of $\mathbf{A}$ consisting of its (entire) rows indexed by $\mathcal{I}$.  ∎

To prove Lemma 5, we will use the following lemma, many variants of which appeared in the literature [e.g., vdV65]. We use the one given by [DWH19a].

**Lemma 8 ([DWH19a])**  *If the rows of random $k \times d$ matrices $\mathbf{A}, \mathbf{B}$ are sampled as an i.i.d. sequence of $k \geq d$ pairs of joint random vectors, then*

$$
k^d\,\mathbb{E}\big[\det(\mathbf{A}^\top \mathbf{B})\big] = k^{\underline{d}}\,\det\big(\mathbb{E}[\mathbf{A}^\top \mathbf{B}]\big). \tag{5}
$$

Here, we use the following standard shorthand: $k^{\underline{d}} = \frac{k!}{(k-d)!} = k\,(k-1)\cdots(k-d+1)$. Note that the above result almost looks like we are claiming that the matrix $\mathbf{A}^\top\mathbf{B}$ is d.p., but in fact it is not because $k^d \neq k^{\underline{d}}$. The difference in those factors is precisely what we are going to correct with the Poisson random variable. We now present the proof of Lemma 5.

**Proof of Lemma 5**  Without loss of generality, it suffices to check Definition 4 with both $\mathcal{I}$ and $\mathcal{J}$ equal $[d]$. We first expand the expectation by conditioning on the value of $K$ and letting $\gamma = \mathbb{E}[K]$:

$$
\begin{aligned}
\mathbb{E}\big[\det(\mathbf{A}^\top \mathbf{B})\big] &= \sum_{k=0}^{\infty} \mathbb{E}\big[\det(\mathbf{A}^\top \mathbf{B}) \mid K=k\big]\,\Pr(K=k) \\
(\text{Lemma 8}) \quad &= \sum_{k=d}^{\infty} \frac{k!\,k^{-d}}{(k-d)!}\,\det\big(\mathbb{E}[\mathbf{A}^\top \mathbf{B} \mid K=k]\big)\frac{\gamma^k \mathrm{e}^{-\gamma}}{k!} \\
&= \sum_{k=d}^{\infty} \left(\frac{\gamma}{k}\right)^d \det\big(\mathbb{E}[\mathbf{A}^\top \mathbf{B} \mid K=k]\big)\frac{\gamma^{k-d}\mathrm{e}^{-\gamma}}{(k-d)!}.
\end{aligned}
$$

Note that $\frac{\gamma}{k}\mathbb{E}[\mathbf{A}^\top\mathbf{B} \mid K=k] = \mathbb{E}[\mathbf{A}^\top\mathbf{B}]$, which is independent of $k$. Thus we can rewrite the above expression as:

$$\det\big(\mathbb{E}[\mathbf{A}^\top\mathbf{B}]\big)\sum_{k=d}^{\infty}\frac{\gamma^{k-d}\mathrm{e}^{-\gamma}}{(k-d)!} = \det\big(\mathbb{E}[\mathbf{A}^\top\mathbf{B}]\big)\sum_{k=0}^{\infty}\frac{\gamma^k\mathrm{e}^{-\gamma}}{k!} = \det\big(\mathbb{E}[\mathbf{A}^\top\mathbf{B}]\big),$$

which concludes the proof. ∎

To prove Lemma 6, we use the following standard determinantal formula which is used to derive the normalization constant of a discrete determinantal point process.

**Lemma 9 ([KT12])** *For any $k \times d$ matrices $\mathbf{A},\mathbf{B}$ we have*

$$\det(\mathbf{I} + \mathbf{A}\mathbf{B}^\top) = \sum_{S\subseteq[k]}\det(\mathbf{A}_{S,*}\mathbf{B}_{S,*}^\top).$$

**Proof of Lemma 6** By Lemma 5, the matrix $\mathbf{B}^\top\mathbf{A}$ is determinant preserving. Applying Lemma 4 we conclude that $\mathbf{I} + \mathbf{B}^\top\mathbf{A}$ is also d.p., so

$$\det\big(\mathbf{I} + \mathbb{E}[\mathbf{B}^\top\mathbf{A}]\big) = \mathbb{E}\big[\det(\mathbf{I} + \mathbf{B}^\top\mathbf{A})\big] = \mathbb{E}\big[\det(\mathbf{I} + \mathbf{A}\mathbf{B}^\top)\big],$$

where the second equality is known as Sylvester's Theorem. We rewrite the expectation of $\det(\mathbf{I} + \mathbf{A}\mathbf{B}^\top)$ by applying Lemma 9. Letting $\gamma = \mathbb{E}[K]$, we obtain:

$$\mathbb{E}\big[\det(\mathbf{I} + \mathbf{A}\mathbf{B}^\top)\big] = \mathbb{E}\bigg[\sum_{S\subseteq[K]}\mathbb{E}\big[\det(\mathbf{A}_{S,*}\mathbf{B}_{S,*}^\top) \mid K\big]\bigg]$$

$$\overset{(*)}{=} \sum_{k=0}^{\infty}\frac{\gamma^k\mathrm{e}^{-\gamma}}{k!}\sum_{i=0}^{k}\binom{k}{i}\mathbb{E}\big[\det(\mathbf{A}\mathbf{B}^\top) \mid K=i\big]$$

$$= \sum_{i=0}^{\infty}\mathbb{E}\big[\det(\mathbf{A}\mathbf{B}^\top) \mid K=i\big]\sum_{k\geq i}^{\infty}\binom{k}{i}\frac{\gamma^k\mathrm{e}^{-\gamma}}{k!}$$

$$= \sum_{i=0}^{\infty}\frac{\gamma^i\mathrm{e}^{-\gamma}}{i!}\mathbb{E}\big[\det(\mathbf{A}\mathbf{B}^\top) \mid K=i\big]\sum_{k\geq i}^{\infty}\frac{\gamma^{k-i}}{(k-i)!} = \mathbb{E}\big[\det(\mathbf{A}\mathbf{B}^\top)\big]\cdot\mathrm{e}^{\gamma},$$

where $(*)$ follows from the exchangeability of the rows of $\mathbf{A}$ and $\mathbf{B}$, which implies that the distribution of $\mathbf{A}_{S,*}\mathbf{B}_{S,*}^\top$ is the same for all subsets $S$ of a fixed size $k$. ∎

## C Proof of Theorem 1

In this section we use $Z_\mu^n$ to denote the normalization constant that appears in (1) when computing an expectation for surrogate design $S_\mu^n$. We first prove Lemma 3.

**Lemma 10 (restated Lemma 3)** *If $\bar{\mathbf{X}} \sim S_\mu^n$ for $n < d$, then we have*

$$\mathbb{E}\big[\mathrm{tr}\big((\bar{\mathbf{X}}^\top\bar{\mathbf{X}})^\dagger\big)\big] = \gamma_n\big(1 - \det\big((\tfrac{1}{\gamma_n}\mathbf{I} + \mathbf{\Sigma}_\mu)^{-1}\mathbf{\Sigma}_\mu\big)\big).$$

**Proof** Let $\mathbf{X} \sim \mu^K$ for $K \sim \mathrm{Poisson}(\gamma_n)$. Note that if $\det(\mathbf{X}\mathbf{X}^\top) > 0$ then using the fact that $\det(\mathbf{A})\mathbf{A}^{-1} = \mathrm{adj}(\mathbf{A})$ for any invertible matrix $\mathbf{A}$, we can write:

$$\det(\mathbf{X}\mathbf{X}^\top)\mathrm{tr}\big((\mathbf{X}^\top\mathbf{X})^\dagger\big) = \det(\mathbf{X}\mathbf{X}^\top)\mathrm{tr}\big((\mathbf{X}\mathbf{X}^\top)^{-1}\big)$$

$$= \mathrm{tr}(\mathrm{adj}(\mathbf{X}\mathbf{X}^\top))$$

$$= \sum_{i=1}^{K}\det(\mathbf{X}_{-i}\mathbf{X}_{-i}^\top),$$

where $\mathbf{X}_{-i}$ is a shorthand for $\mathbf{X}_{[K]\setminus\{i\},*}$. Assumption 2 ensures that $\Pr\{\det(\mathbf{X}\mathbf{X}^\top) > 0\} = 1$, which allows us to write:

$$
\begin{aligned}
Z_\mu^n \cdot \mathbb{E}\big[\mathrm{tr}\big((\bar{\mathbf{X}}^\top\bar{\mathbf{X}})^\dagger\big)\big] &= \mathbb{E}\bigg[\sum_{i=1}^K \det(\mathbf{X}_{-i}\mathbf{X}_{-i}^\top) \,\big|\, \det(\mathbf{X}\mathbf{X}^\top) > 0\bigg] \cdot \overbrace{\Pr\{\det(\mathbf{X}\mathbf{X}^\top) > 0\}}^{1} \\
&= \sum_{k=0}^d \frac{\gamma_n^k e^{-\gamma_n}}{k!} \mathbb{E}\bigg[\sum_{i=1}^k \det(\mathbf{X}_{-i}\mathbf{X}_{-i}^\top) \,\big|\, K = k\bigg] \\
&= \sum_{k=0}^d \frac{\gamma_n^k e^{-\gamma_n}}{k!}\, k\, \mathbb{E}\big[\det(\mathbf{X}\mathbf{X}^\top) \,\big|\, K = k-1\big] \\
&= \gamma_n \sum_{k=0}^{d-1} \frac{\gamma_n^k e^{-\gamma_n}}{k!} \mathbb{E}\big[\det(\mathbf{X}\mathbf{X}^\top) \,\big|\, K = k\big] \\
&= \gamma_n\Big(\mathbb{E}\big[\det(\mathbf{X}\mathbf{X}^\top)\big] - \frac{\gamma_n^d e^{-\gamma_n}}{d!}\mathbb{E}\big[\det(\mathbf{X})^2 \,\big|\, K = d\big]\Big) \\
&\overset{(*)}{=} \gamma_n\big(e^{-\gamma_n}\det(\mathbf{I} + \gamma_n\boldsymbol{\Sigma}_\mu) - e^{-\gamma_n}\det(\gamma_n\boldsymbol{\Sigma}_\mu)\big),
\end{aligned}
$$

where $(*)$ uses Lemma 6 for the first term and Lemma 8 for the second term. We obtain the desired result by dividing both sides by $Z_\mu^n = e^{-\gamma_n}\det(\mathbf{I} + \gamma_n\boldsymbol{\Sigma}_\mu)$. ∎

In the over-determined regime, a more general matrix expectation formula can be shown (omitting the trace). The following result is related to an expectation formula derived by [DWH19b], however they use a slightly different determinantal design so the results are incomparable.

**Lemma 11** *If $\bar{\mathbf{X}} \sim S_\mu^n$ and $n > d$, then we have*

$$
\mathbb{E}\big[(\bar{\mathbf{X}}^\top\bar{\mathbf{X}})^\dagger\big] = \boldsymbol{\Sigma}_\mu^{-1} \cdot \frac{1 - e^{-\gamma_n}}{\gamma_n}.
$$

**Proof** Let $\mathbf{X} \sim \mu^K$ for $K \sim \mathrm{Poisson}(\gamma_n)$. Assumption 2 implies that for $K \neq d-1$ we have

$$
\det(\mathbf{X}^\top\mathbf{X})(\mathbf{X}^\top\mathbf{X})^\dagger = \mathrm{adj}(\mathbf{X}^\top\mathbf{X}), \tag{6}
$$

however when $k = d-1$ then (6) does not hold because $\det(\mathbf{X}^\top\mathbf{X}) = 0$ while $\mathrm{adj}(\mathbf{X}^\top\mathbf{X})$ may be non-zero. It follows that:

$$
\begin{aligned}
Z_\mu^n \cdot \mathbb{E}\big[(\bar{\mathbf{X}}^\top\bar{\mathbf{X}})^\dagger\big] &= \mathbb{E}\big[\det(\mathbf{X}^\top\mathbf{X})(\mathbf{X}^\top\mathbf{X})^\dagger\big] \\
&= \mathbb{E}\big[\mathrm{adj}(\mathbf{X}^\top\mathbf{X})\big] - \frac{\gamma_n^{d-1} e^{-\gamma_n}}{(d-1)!}\mathbb{E}\big[\mathrm{adj}(\mathbf{X}^\top\mathbf{X}) \,\big|\, K = d-1\big] \\
&\overset{(*)}{=} \mathrm{adj}\big(\mathbb{E}[\mathbf{X}^\top\mathbf{X}]\big) - \frac{\gamma_n^{d-1} e^{-\gamma_n}}{(d-1)^{d-1}}\mathrm{adj}\big(\mathbb{E}[\mathbf{X}^\top\mathbf{X} \,|\, K = d-1]\big) \\
&= \mathrm{adj}(\gamma_n\boldsymbol{\Sigma}_\mu) - e^{-\gamma_n}\mathrm{adj}(\gamma_n\boldsymbol{\Sigma}_\mu) \\
&= \det(\gamma_n\boldsymbol{\Sigma}_\mu)\,(\gamma_n\boldsymbol{\Sigma}_\mu)^{-1}(1 - e^{-\gamma_n}) \\
&= \det(\gamma_n\boldsymbol{\Sigma}_\mu)\,\boldsymbol{\Sigma}_\mu^{-1} \cdot \frac{1 - e^{-\gamma_n}}{\gamma_n},
\end{aligned}
$$

where the first term in $(*)$ follows from Lemma 6 and (4), whereas the second term comes from Lemma 2.3 of [DWH19b]. Dividing both sides by $Z_\mu^n = \det(\gamma_n\boldsymbol{\Sigma}_\mu)$ completes the proof. ∎

Applying the closed form expressions from Lemmas 2, 3 and 11, we derive the formula for the MSE and prove Theorem 1 (we defer the proof of Lemma 2 to Appendix D).

**Proof of Theorem 1** First, assume that $n < d$, in which case we have $\gamma_n = \frac{1}{\lambda_n}$ and moreover

$$
\begin{aligned}
n &= \mathrm{tr}\big(\boldsymbol{\Sigma}_\mu(\boldsymbol{\Sigma}_\mu + \lambda_n\mathbf{I})^{-1}\big) \\
&= \mathrm{tr}\big((\boldsymbol{\Sigma}_\mu + \lambda_n\mathbf{I} - \lambda_n\mathbf{I})(\boldsymbol{\Sigma}_\mu + \lambda_n\mathbf{I})^{-1}\big) \\
&= d - \lambda_n\mathrm{tr}\big((\boldsymbol{\Sigma}_\mu + \lambda_n\mathbf{I})^{-1}\big),
\end{aligned}
$$

so we can write $\lambda_n$ as $(d-n)/\mathrm{tr}((\boldsymbol{\Sigma}_\mu + \lambda_n\mathbf{I})^{-1})$. From this and Lemmas 2 and 10, we obtain the desired expression, where recall that $\alpha_n = \det\!\big(\boldsymbol{\Sigma}_\mu(\boldsymbol{\Sigma}_\mu + \tfrac{1}{\gamma_n})^{-1}\big)$:

$$
\begin{aligned}
\mathrm{MSE}\big[\bar{\mathbf{X}}^\dagger \bar{\mathbf{y}}\big] &= \sigma^2\,\gamma_n(1-\alpha_n) + \tfrac{1}{\gamma_n}\,\mathbf{w}^{*\top}(\boldsymbol{\Sigma}_\mu + \tfrac{1}{\gamma_n}\mathbf{I})^{-1}\mathbf{w}^* \\
&\overset{(a)}{=} \sigma^2\,\frac{1-\alpha_n}{\lambda_n} + \lambda_n\,\mathbf{w}^{*\top}(\boldsymbol{\Sigma}_\mu + \lambda_n\mathbf{I})^{-1}\mathbf{w}^* \\
&\overset{(b)}{=} \sigma^2\mathrm{tr}\big((\boldsymbol{\Sigma}_\mu + \lambda_n\mathbf{I})^{-1}\big)\frac{1-\alpha_n}{d-n} + (d-n)\frac{\mathbf{w}^{*\top}(\boldsymbol{\Sigma}_\mu + \lambda_n\mathbf{I})^{-1}\mathbf{w}^*}{\mathrm{tr}\big((\boldsymbol{\Sigma}_\mu + \lambda_n\mathbf{I})^{-1}\big)}.
\end{aligned}
$$

While the expression given after $(a)$ is simpler than the one after $(b)$, the latter better illustrates how the MSE depends on the sample size $n$ and the dimension $d$. Now, assume that $n > d$. In this case, we have $\gamma_n = n-d$ and apply Lemma 11:

$$
\mathrm{MSE}\big[\bar{\mathbf{X}}^\dagger\bar{\mathbf{y}}\big] = \sigma^2\,\mathrm{tr}(\boldsymbol{\Sigma}_\mu^{-1})\,\frac{1-\mathrm{e}^{-\gamma_n}}{\gamma_n} = \sigma^2\,\mathrm{tr}(\boldsymbol{\Sigma}_\mu^{-1})\,\frac{1-\beta_n}{n-d}.
$$

The case of $n = d$ was shown in Theorem 2.12 of [DWH19b]. This concludes the proof. ∎

## D  Proof of Theorem 2

As in the previous section, we use $Z_\mu^n$ to denote the normalization constant that appears in (1) when computing an expectation for surrogate design $S_\mu^n$. Recall that our goal is to compute the expected value of $\bar{\mathbf{X}}^\dagger\bar{\mathbf{y}}$ under the surrogate design $S_\mu^n$. Similarly as for Theorem 1, the case of $n = d$ was shown in Theorem 2.10 of [DWH19b]. We break the rest down into the under-determined case ($n < d$) and the over-determined case ($n > d$), starting with the former. Recall that we do *not* require any modeling assumptions on the responses.

**Lemma 12** *If $\bar{\mathbf{X}} \sim S_\mu^n$ and $n < d$, then for any $y(\cdot)$ such that $\mathbb{E}_{\mu,y}[y(\mathbf{x})\,\mathbf{x}]$ is well-defined, denoting $\bar{y}_i$ as $y(\bar{\mathbf{x}}_i)$, we have*

$$
\mathbb{E}\big[\bar{\mathbf{X}}^\dagger\bar{\mathbf{y}}\big] = \big(\boldsymbol{\Sigma}_\mu + \tfrac{1}{\gamma_n}\mathbf{I}\big)^{-1}\mathbb{E}_{\mu,y}[y(\mathbf{x})\,\mathbf{x}].
$$

**Proof** Let $\mathbf{X} \sim \mu^K$ for $K \sim \mathrm{Poisson}(\gamma_n)$ and denote $y(\mathbf{x}_i)$ as $y_i$. Note that when $\det(\mathbf{X}\mathbf{X}^\top) > 0$, then the $j$th entry of $\mathbf{X}^\dagger\mathbf{y}$ equals $\mathbf{f}_j^\top(\mathbf{X}\mathbf{X}^\top)^{-1}\mathbf{y}$, where $\mathbf{f}_j$ is the $j$th column of $\mathbf{X}$, so:

$$
\begin{aligned}
\det(\mathbf{X}\mathbf{X}^\top)\,(\mathbf{X}^\dagger\mathbf{y})_j &= \det(\mathbf{X}\mathbf{X}^\top)\,\mathbf{f}_j^\top(\mathbf{X}\mathbf{X}^\top)^{-1}\mathbf{y} \\
&= \det(\mathbf{X}\mathbf{X}^\top + \mathbf{y}\mathbf{f}_j^\top) - \det(\mathbf{X}\mathbf{X}^\top).
\end{aligned}
$$

If $\det(\mathbf{X}\mathbf{X}^\top) = 0$, then also $\det(\mathbf{X}\mathbf{X}^\top + \mathbf{y}\mathbf{f}_j^\top) = 0$, so we can write:

$$
\begin{aligned}
Z_\mu^n \cdot \mathbb{E}\big[(\bar{\mathbf{X}}^\dagger\bar{\mathbf{y}})_j\big] &= \mathbb{E}\big[\det(\mathbf{X}\mathbf{X}^\top)(\mathbf{X}^\dagger\mathbf{y})_j\big] \\
&= \mathbb{E}\big[\det(\mathbf{X}\mathbf{X}^\top + \mathbf{y}\mathbf{f}_j^\top) - \det(\mathbf{X}\mathbf{X}^\top)\big] \\
&= \mathbb{E}\big[\det\big([\mathbf{X},\mathbf{y}][\mathbf{X},\mathbf{f}_j]^\top\big)\big] - \mathbb{E}\big[\det(\mathbf{X}\mathbf{X}^\top)\big] \\
&\overset{(a)}{=} \mathrm{e}^{-\gamma_n}\det\!\left(\mathbf{I} + \gamma_n\,\mathbb{E}_{\mu,y}\!\left[\begin{pmatrix} \mathbf{x}\mathbf{x}^\top & \mathbf{x}\,y(\mathbf{x}) \\ x_j\,\mathbf{x}^\top & x_j\,y(\mathbf{x}) \end{pmatrix}\right]\right) - \mathrm{e}^{-\gamma_n}\det(\mathbf{I} + \gamma_n\boldsymbol{\Sigma}_\mu) \\
&\overset{(b)}{=} \mathrm{e}^{-\gamma_n}\det(\mathbf{I} + \gamma_n\boldsymbol{\Sigma}_\mu) \\
&\quad \times \Big(\mathbb{E}_{\mu,y}\big[\gamma_n x_j\,y(\mathbf{x})\big] - \mathbb{E}_\mu\big[\gamma_n x_j\,\mathbf{x}^\top\big](\mathbf{I} + \gamma_n\boldsymbol{\Sigma}_\mu)^{-1}\mathbb{E}_{\mu,y}\big[\gamma_n\mathbf{x}\,y(\mathbf{x})\big]\Big),
\end{aligned}
$$

where $(a)$ uses Lemma 6 twice, with the first application involving two different matrices $\mathbf{A} = [\mathbf{X},\mathbf{y}]$ and $\mathbf{B} = [\mathbf{X},\mathbf{f}_j]$, whereas $(b)$ is a standard determinantal identity [see Fact 2.14.2 in Ber11]. Dividing both sides by $Z_\mu^n$ and letting $\mathbf{v}_{\mu,y} = \mathbb{E}_{\mu,y}[y(\mathbf{x})\,\mathbf{x}]$, we obtain that:

$$
\begin{aligned}
\mathbb{E}\big[\bar{\mathbf{X}}^\dagger\bar{\mathbf{y}}\big] &= \gamma_n\mathbf{v}_{\mu,y} - \gamma_n^2\boldsymbol{\Sigma}_\mu(\mathbf{I} + \gamma_n\boldsymbol{\Sigma}_\mu)^{-1}\mathbf{v}_{\mu,y} \\
&= \gamma_n\big(\mathbf{I} - \gamma_n\boldsymbol{\Sigma}_\mu(\mathbf{I} + \gamma_n\boldsymbol{\Sigma}_\mu)^{-1}\big)\mathbf{v}_{\mu,y} = \gamma_n(\mathbf{I} + \gamma_n\boldsymbol{\Sigma}_\mu)^{-1}\mathbf{v}_{\mu,y},
\end{aligned}
$$

which completes the proof. ∎

We return to Lemma 2, regarding the expected orthogonal projection onto the complement of the row-span of $\bar{\mathbf{X}}$, i.e., $\mathbb{E}[\mathbf{I} - \bar{\mathbf{X}}^\dagger \bar{\mathbf{X}}]$, which follows as a corollary of Lemma 12.

**Proof of Lemma 2** We let $y(\mathbf{x}) = x_j$ where $j \in [d]$ and apply Lemma 12 for each $j$, obtaining:

$$\mathbf{I} - \mathbb{E}\big[\bar{\mathbf{X}}^\dagger \bar{\mathbf{X}}\big] = \mathbf{I} - (\boldsymbol{\Sigma}_\mu + \tfrac{1}{\gamma_n}\mathbf{I})^{-1}\boldsymbol{\Sigma}_\mu,$$

from which the result follows by simple algebraic manipulation. ∎

We move on to the over-determined case, where the ridge regularization of adding the identity to $\boldsymbol{\Sigma}_\mu$ vanishes. Recall that we assume throughout the paper that $\boldsymbol{\Sigma}_\mu$ is invertible.

**Lemma 13** *If $\bar{\mathbf{X}} \sim S_\mu^n$ and $n > d$, then for any real-valued random function $y(\cdot)$ such that $\mathbb{E}_{\mu,y}[y(\mathbf{x})\,\mathbf{x}]$ is well-defined, denoting $\bar{y}_i$ as $y(\bar{\mathbf{x}}_i)$, we have*

$$\mathbb{E}\big[\bar{\mathbf{X}}^\dagger \bar{\mathbf{y}}\big] = \boldsymbol{\Sigma}_\mu^{-1}\mathbb{E}_{\mu,y}\big[y(\mathbf{x})\,\mathbf{x}\big].$$

**Proof** Let $\mathbf{X} \sim \mu^K$ for $K \sim \mathrm{Poisson}(\gamma_n)$ and denote $y_i = y(\mathbf{x}_i)$. Similarly as in the proof of Lemma 12, we note that when $\det(\mathbf{X}^\top \mathbf{X}) > 0$, then the $j$th entry of $\mathbf{X}^\dagger \mathbf{y}$ equals $\mathbf{e}_j^\top (\mathbf{X}^\top \mathbf{X})^{-1}\mathbf{X}^\top \mathbf{y}$, where $\mathbf{e}_j$ is the $j$th standard basis vector, so:

$$\det(\mathbf{X}^\top \mathbf{X})\,(\mathbf{X}^\dagger \mathbf{y})_j = \det(\mathbf{X}^\top \mathbf{X})\,\mathbf{e}_j^\top(\mathbf{X}^\top \mathbf{X})^{-1}\mathbf{X}^\top \mathbf{y} = \det(\mathbf{X}^\top \mathbf{X} + \mathbf{X}^\top \mathbf{y}\mathbf{e}_j^\top) - \det(\mathbf{X}^\top \mathbf{X}).$$

If $\det(\mathbf{X}^\top \mathbf{X}) = 0$, then also $\det(\mathbf{X}^\top \mathbf{X} + \mathbf{X}^\top \mathbf{y}\mathbf{e}_j^\top) = 0$. We proceed to compute the expectation:

$$
\begin{aligned}
Z_\mu^n \cdot \mathbb{E}\big[(\bar{\mathbf{X}}^\dagger \bar{\mathbf{y}})_j\big] &= \mathbb{E}\big[\det(\mathbf{X}^\top \mathbf{X})(\mathbf{X}^\dagger \mathbf{y})_j\big]\\
&= \mathbb{E}\big[\det(\mathbf{X}^\top \mathbf{X} + \mathbf{X}^\top \mathbf{y}\mathbf{e}_j^\top) - \det(\mathbf{X}^\top \mathbf{X})\big]\\
&= \mathbb{E}\big[\det\big(\mathbf{X}^\top(\mathbf{X} + \mathbf{y}\mathbf{e}_j^\top)\big)\big] - \mathbb{E}\big[\det(\mathbf{X}^\top \mathbf{X})\big]\\
&\overset{(*)}{=} \det\Big(\gamma_n\,\mathbb{E}_{\mu,y}\big[\mathbf{x}(\mathbf{x} + y(\mathbf{x})\mathbf{e}_j)^\top\big]\Big) - \det(\gamma_n\boldsymbol{\Sigma}_\mu)\\
&= \det\big(\gamma_n\boldsymbol{\Sigma}_\mu + \gamma_n\mathbb{E}_{\mu,y}[\mathbf{x}\,y(\mathbf{x})]\mathbf{e}_j^\top\big) - \det(\gamma_n\boldsymbol{\Sigma}_\mu)\\
&= \det(\gamma_n\boldsymbol{\Sigma}_\mu)\cdot \gamma_n\mathbf{e}_j^\top(\gamma_n\boldsymbol{\Sigma}_\mu)^{-1}\mathbb{E}_{\mu,y}\big[y(\mathbf{x})\,\mathbf{x}\big],
\end{aligned}
$$

where $(*)$ uses Lemma 5 twice (the first time, with $\mathbf{A} = \mathbf{X}$ and $\mathbf{B} = \mathbf{X} + \mathbf{y}\mathbf{e}_j^\top$). Dividing both sides by $Z_\mu^n = \det(\gamma_n\boldsymbol{\Sigma}_\mu)$ concludes the proof. ∎

We combine Lemmas 12 and 13 to obtain the proof of Theorem 2.

**Proof of Theorem 2** The case of $n = d$ follows directly from Theorem 2.10 of [DWH19a]. Assume that $n < d$. Then we have $\gamma_n = \frac{1}{\lambda_n}$, so the result follows from Lemma 12. If $n > d$, then the result follows from Lemma 13. ∎

## E   Proof of Theorem 3

The proof of Theorem 3 follows the standard decomposition of MSE in Equation 2, and in the process, establishes consistency of the variance and bias terms independently. To this end, we introduce the following two useful lemmas that capture the limiting behavior of the variance and bias terms, respectively.

**Lemma 14** *Under the setting of Theorem 3, we have, as $n, d \to \infty$ with $n/d \to \bar{c} \in (0, \infty) \setminus \{1\}$ that*

$$
\begin{cases}
\mathbb{E}\big[\mathrm{tr}((\mathbf{X}^\top \mathbf{X})^\dagger)\big] - (1 - \alpha_n)\lambda_n^{-1} \to 0, & \text{for } \bar{c} < 1,\\
\mathbb{E}\big[\mathrm{tr}((\mathbf{X}^\top \mathbf{X})^\dagger)\big] - \frac{1 - \beta_n}{n - d}\cdot \mathrm{tr}\boldsymbol{\Sigma}^{-1} \to 0, & \text{for } \bar{c} > 1
\end{cases}
\tag{7}
$$

*where $\lambda_n \geq 0$ is the unique solution to $n = \mathrm{tr}(\boldsymbol{\Sigma}(\boldsymbol{\Sigma} + \lambda_n\mathbf{I})^{-1})$, $\alpha_n = \det(\boldsymbol{\Sigma}(\boldsymbol{\Sigma} + \lambda_n\mathbf{I})^{-1})$, and $\beta_n = e^{d-n}$.*

The second term in the MSE derivation (2), $\mathbb{E}[\mathbf{I} - \mathbf{X}^{\dagger}\mathbf{X}]$, involves the expectation of a projection onto the orthogonal complement of a sub-Gaussian general position sample $\mathbf{X}$. This term is zero when $n > d$, and for $n < d$ we prove in appendix E.2 that the surrogate design's bias $\mathcal{B}(\mathbf{\Sigma}, n)$ provides an asymptotically consistent approximation to all of the eigenvectors and eigenvalues:

**Lemma 15** *Under the setting of Theorem 3, for* $\mathbf{w} \in \mathbb{R}^d$ *of bounded Euclidean norm (i.e.,* $\|\mathbf{w}\| \leq C'$ *for all d), we have, as* $n, d \to \infty$ *with* $n/d \to \bar{c} \in (0, 1)$ *that*

$$\mathbf{w}^{\top}\mathbb{E}[\mathbf{I} - \mathbf{X}^{\dagger}\mathbf{X}]\mathbf{w} - \lambda_n \mathbf{w}^{\top}(\mathbf{\Sigma} + \lambda_n \mathbf{I})^{-1}\mathbf{w} \to 0 \tag{8}$$

*while* $\mathbf{I} - \mathbf{X}^{\dagger}\mathbf{X} = 0$ *for* $\bar{c} > 1$.

### E.1 Proof of lemma 14

#### E.1.1 The $\bar{c} \in (0, 1)$ case

For $n < d$, we first establish (1) $\liminf_n \lambda_n > 0$ and (2) $\alpha_n \to 0$. To prove (1), by hypothesis $\mathbf{\Sigma} \succeq c\mathbf{I}$ for all $d$. Since $\frac{n}{d} < 1$, we have (by definition of $\lambda_n$) for some $\delta > 0$

$$1 - \delta > \frac{n}{d} = \frac{1}{d}\operatorname{tr}(\mathbf{\Sigma}(\mathbf{\Sigma} + \lambda_n\mathbf{I})^{-1}) > \frac{c}{c + \lambda_n}$$

Rearranging, we have $\lambda_n > \frac{\delta c}{1 - \delta} > 0$. For (2), let $(\tau_i)_{i \in [d]}$ denote the eigenvalues of $\mathbf{\Sigma}$. Since $1 - x \leq e^{-x}$ and $C\mathbf{I} \succeq \mathbf{\Sigma} \succeq c\mathbf{I}$ for all $d$,

$$\alpha_n = \prod_{i=1}^{d} \frac{\tau_i}{\tau_i + \lambda_n} \leq \left(\frac{C}{C + \lambda_n}\right)^d = \left(1 - \frac{\lambda_n}{C + \lambda_n}\right)^d \leq \exp\left(-d\frac{\lambda_n}{C + \lambda_n}\right)$$

and since $\lambda_n > 0$ eventually as $d \to \infty$ we have $\alpha_n \to 0$ so that $(1 - \alpha_n)\lambda_n^{-1} - \lambda_n^{-1} \to 0$.

As a consequence of (2) and Slutsky's theorem, it suffices to show $\operatorname{tr}(\mathbf{X}^{\top}\mathbf{X})^{\dagger} - \lambda_n^{-1} \xrightarrow{d} 0$ as $n, d \to \infty$. To do this, we consider the limiting behavior of $\operatorname{tr}(\mathbf{X}^{\top}\mathbf{X})^{\dagger}/n = \operatorname{tr}(\mathbf{X}\mathbf{X}^{\top})^{\dagger}/n$ as $n/d \to \bar{c} \in (0, 1)$, for $\mathbf{X} = \mathbf{Z}\mathbf{\Sigma}^{\frac{1}{2}}$ with $\mathbf{Z} \in \mathbb{R}^{n \times d}$ having i.i.d. zero mean, unit variance sub-Gaussian entries, i.e., the behavior of

$$\lim_{n,d\to\infty} \lim_{z\to 0^+} \frac{1}{n}\operatorname{tr}\left(\frac{1}{n}\mathbf{X}\mathbf{X}^{\top} + z\mathbf{I}_n\right)^{-1} \tag{9}$$

by definition of the pseudo-inverse.

The proof comes in three steps: (i) for fixed $z > 0$, consider the limiting behavior of $\delta(z) \equiv \operatorname{tr}(\mathbf{X}\mathbf{X}^{\top}/n + z\mathbf{I}_n)^{-1}/n$ as $n, d \to \infty$ and state

$$\lim_{n,d\to\infty} \delta(z) - m(z) \to 0 \tag{10}$$

almost surely for some $m(z)$ to be defined; (ii) show that both $\delta(z)$ and its derivate $\delta'(z)$ are uniformly bounded (by some quantity independent of $z > 0$) so that by Arzela-Ascoli theorem, $\delta(z)$ converges uniformly to its limit and we are allowed to take $z \to 0^+$ in (10) and state

$$\lim_{z\to 0^+} \lim_{n,d\to\infty} \delta(z) - \lim_{z\to 0^+} m(z) \to 0 \tag{11}$$

almost surely, given that the limit $\lim_{z\to 0^+} m(z) \equiv m(0)$ exists and eventually (iii) exchange the two limits in (11) with Moore-Osgood theorem, to reach

$$\lim_{n,d\to\infty} \lim_{z\to 0^+} \frac{1}{n}\operatorname{tr}\left(\frac{1}{n}\mathbf{X}\mathbf{X}^{\top} + z\mathbf{I}_n\right)^{-1} - m(0) \to 0.$$

Step (i) follows from [SB95] that, we have, for $z > 0$ that

$$\delta(z) \equiv \frac{1}{n}\operatorname{tr}\left(\frac{1}{n}\mathbf{X}\mathbf{X}^{\top} + z\mathbf{I}_n\right)^{-1} - m(z) \to 0$$

almost surely as $n, d \to \infty$, for $m(z)$ the unique positive solution to

$$m(z) = \left( z + \frac{1}{n}\text{tr}\boldsymbol{\Sigma}(\mathbf{I} + m(z)\boldsymbol{\Sigma})^{-1} \right)^{-1}. \tag{12}$$

For the above step (ii), we use the assumption $\boldsymbol{\Sigma} \succeq c\mathbf{I} \succ 0$ for all $d$ large, so that with $\mathbf{X} = \mathbf{Z}\boldsymbol{\Sigma}^{\frac{1}{2}}$, we have for large enough $n, d$ that

$$\lambda_{\min}(\mathbf{X}\mathbf{X}^\top/n) \geq \lambda_{\min}(\mathbf{Z}\mathbf{Z}^\top/n)\lambda_{\min}(\boldsymbol{\Sigma}) \geq \frac{c}{2}(\sqrt{\bar{c}} - 1)^2$$

almost surely, where we used Bai-Yin theorem [BY$^+$93], which states that the minimum eigenvalue of $\mathbf{Z}\mathbf{Z}^\top/n$ is almost surely larger than $(\sqrt{\bar{c}} - 1)^2/2$ for $n < d$ sufficiently large. Note that here the case $\bar{c} = 1$ is excluded.

Observe that

$$|\delta(z)| = \left| \frac{1}{n}\text{tr}\left( \frac{1}{n}\mathbf{X}\mathbf{X}^\top + z\mathbf{I}_n \right)^{-1} \right| \leq \frac{1}{\lambda_{\min}(\mathbf{X}\mathbf{X}^\top/n)}$$

and similarly for its derivative, so that we are allowed to take the $z \to 0^+$ limit. Note that the existence of the $\lim_{z\to 0^+} m(z)$ for $m(z)$ defined in (12) is well known, see for example [LP11]. Then, by Moore-Osgood theorem we finish step (iii) and by concluding that

$$\text{tr}(\mathbf{X}^\top\mathbf{X})^\dagger - m(0) \to 0$$

for $m(0) = \lambda_n^{-1}$ the unique solution to $\lambda_n^{-1} = \left(\frac{1}{n}\text{tr}\boldsymbol{\Sigma}(\mathbf{I} + \lambda_n^{-1}\boldsymbol{\Sigma})^{-1}\right)^{-1}$, or equivalently, to

$$n = \text{tr}\boldsymbol{\Sigma}(\boldsymbol{\Sigma} + \lambda_n\mathbf{I})^{-1}$$

as desired.

### E.1.2 The $\bar{c} \in (1, \infty)$ case

First note that as $n, d \to \infty$ with $n > d$, we have $\beta_n = e^{d-n} \to 0$ and it it suffices to show

$$\text{tr}(\mathbf{X}^\top\mathbf{X})^\dagger - \frac{1}{n-d}\text{tr}\boldsymbol{\Sigma}^{-1} \to 0$$

almost surely to conclude the proof.

In the $\bar{c} \in (1, \infty)$ case, it is more convenient to work on the following co-resolvent

$$\lim_{n,d\to\infty} \lim_{z\to 0^+} \frac{1}{n}\text{tr}\left( \frac{1}{n}\mathbf{X}^\top\mathbf{X} + z\mathbf{I}_d \right)^{-1}$$

where we recall $\mathbf{X}^\top\mathbf{X} = \boldsymbol{\Sigma}^{\frac{1}{2}}\mathbf{Z}^\top\mathbf{Z}\boldsymbol{\Sigma}^{\frac{1}{2}} \in \mathbb{R}^{d\times d}$ and following the same three-step procedure as in the $\bar{c} < 1$ case above. The only difference is in step (i) we need to assess the asymptotic behavior of $\delta \equiv \text{tr}(\mathbf{X}^\top\mathbf{X}/n + z\mathbf{I}_d)^{-1}/n$. This was established in [BS$^+$98] where it was shown that, for $z > 0$ we have

$$\frac{1}{n}\text{tr}(\mathbf{X}^\top\mathbf{X}/n + z\mathbf{I}_d)^{-1} - \frac{d}{n}m(z) \to 0$$

almost surely as $n, d \to \infty$, for $m(z)$ the unique solution to

$$m(z) = \frac{1}{d}\text{tr}\left( \left(1 - \frac{d}{n} - \frac{d}{n}zm(z)\right)\boldsymbol{\Sigma} - z\mathbf{I}_d \right)^{-1}$$

so that for $d < n$ by taking $z = 0$ we have

$$m(0) = \frac{n}{d}\frac{1}{n-d}\text{tr}\boldsymbol{\Sigma}^{-1}.$$

The steps (ii) and (iii) follow exactly the same line of arguments as the $\bar{c} < 1$ case and are thus omitted.

### E.2 Proof of lemma 15

Since $\mathbf{X}^\dagger\mathbf{X} = \mathbf{X}^\top(\mathbf{X}\mathbf{X}^\top)^\dagger\mathbf{X}$, to prove lemma 15, we are interested in the limiting behavior of the following quadratic form

$$\lim_{n,d\to\infty}\lim_{z\to 0^+}\frac{1}{n}\mathbf{w}^\top\mathbf{X}^\top\left(\frac{1}{n}\mathbf{X}\mathbf{X}^\top + z\mathbf{I}_n\right)^{-1}\mathbf{X}\mathbf{w}$$

for deterministic $\mathbf{w}\in\mathbb{R}^d$ of bounded Euclidean norm (i.e., $\|\mathbf{w}\| \le C'$ as $n,d\to\infty$), as $n,d\to\infty$ with $n/d\to\bar{c}\in(0,1)$. The limiting behavior of the above quadratic form, or more generally, bilinear form of the type $\frac{1}{n}\mathbf{w}_1^\top\mathbf{X}^\top\left(\frac{1}{n}\mathbf{X}\mathbf{X}^\top + z\mathbf{I}_n\right)^{-1}\mathbf{X}\mathbf{w}_2$ for $\mathbf{w}_1,\mathbf{w}_2\in\mathbb{R}^d$ of bounded Euclidean norm are widely studied in random matrix literature, see for example [HLNV13].

For the proof of Lemma 15 we follow the same protocol as that of Lemma 14, namely: (i) we consider, for fixed $z > 0$, the limiting behavior of $\frac{1}{n}\mathbf{w}^\top\mathbf{X}^\top\left(\frac{1}{n}\mathbf{X}\mathbf{X}^\top + z\mathbf{I}_n\right)^{-1}\mathbf{X}\mathbf{w}$. Note that

$$\delta(z) \equiv \frac{1}{n}\mathbf{w}^\top\mathbf{X}^\top\left(\frac{1}{n}\mathbf{X}\mathbf{X}^\top + z\mathbf{I}_n\right)^{-1}\mathbf{X}\mathbf{w} = \mathbf{w}^\top\left(\frac{1}{n}\mathbf{X}^\top\mathbf{X} + z\mathbf{I}_d\right)^{-1}\frac{1}{n}\mathbf{X}^\top\mathbf{X}\mathbf{w}$$

$$= \|\mathbf{w}\|^2 - z\mathbf{w}^\top\left(\frac{1}{n}\mathbf{X}^\top\mathbf{X} + z\mathbf{I}_d\right)^{-1}\mathbf{w}$$

and it remains to work on the second $z\mathbf{w}^\top\left(\frac{1}{n}\mathbf{X}^\top\mathbf{X} + z\mathbf{I}_d\right)^{-1}\mathbf{w}$ term. It follows from [HLNV13] that

$$z\mathbf{w}^\top\left(\frac{1}{n}\mathbf{X}^\top\mathbf{X} + z\mathbf{I}_d\right)^{-1}\mathbf{w} - \mathbf{w}^\top(\mathbf{I}_d + m(z)\mathbf{\Sigma})^{-1}\mathbf{w}^\top \to 0$$

almost surely as $n,d\to\infty$, where we recall $m(z)$ is the unique solution to (12).

We move on to step (ii), under the assumption that $c \le \lambda_{\min}(\mathbf{\Sigma}) \le \lambda_{\max}(\mathbf{\Sigma}) \le C$ and $\|\mathbf{w}\| \le C'$, we have

$$\lambda_{\max}\left(\frac{1}{n}\mathbf{X}^\top\left(\frac{1}{n}\mathbf{X}\mathbf{X}^\top + z\mathbf{I}_n\right)^{-1}\mathbf{X}\right) \le \frac{\lambda_{\max}(\mathbf{X}\mathbf{X}^\top/n)}{\lambda_{\min}(\mathbf{X}\mathbf{X}^\top/n) + z} \le \frac{\lambda_{\max}(\mathbf{Z}\mathbf{Z}^\top/n)\lambda_{\max}(\mathbf{\Sigma})}{\lambda_{\min}(\mathbf{Z}\mathbf{Z}^\top/n)\lambda_{\min}(\mathbf{\Sigma})}$$

$$\le 4\frac{(\sqrt{\bar{c}}+1)^2 C}{(\sqrt{\bar{c}}-1)^2 c}$$

so that $\delta(z)$ remains bounded and similarly for its derivative $\delta'(z)$, which, by Arzela-Ascoli theorem, yields uniform convergence and we are allowed to take the $z\to 0^+$ limit. Ultimately, in step (iii) we exchange the two limits with Moore-Osgood theorem, concluding the proof.

### E.3 Finishing the proof of Theorem 3

To finish the proof of Theorem 3, it remains to write

$$\mathrm{MSE}\big[\mathbf{X}^\dagger\mathbf{y}\big] = \sigma^2\mathbb{E}\big[\mathrm{tr}\big((\mathbf{X}^\top\mathbf{X})^\dagger\big)\big] + \mathbf{w}^{*\top}\mathbb{E}\big[\mathbf{I} - \mathbf{X}^\dagger\mathbf{X}\big]\mathbf{w}^*$$

Since $\lambda_n = \frac{d-n}{\mathrm{tr}(\mathbf{\Sigma}+\lambda_n\mathbf{I})^{-1}}$, by Lemma 14 and Lemma 15 we have $\mathrm{MSE}\big[\mathbf{X}^\dagger\mathbf{y}\big] - \mathcal{M}(\mathbf{\Sigma},\mathbf{w}^*,\sigma^2,n) \to 0$ as $n,d\to\infty$ with $n/d\to\bar{c}\in(0,\infty)\setminus\{1\}$, which concludes the proof of Theorem 3.

## F  Additional details for empirical evaluation

Our empirical investigation of the rate of asymptotic convergence in Theorem 3 (and, more specifically, the variance and bias discrepancies defined in Section 5), in the context of Gaussian random matrices, is related to open problems which have been extensively studied in the literature. Note that when $\mathbf{X} = \mathbf{Z}\mathbf{\Sigma}^{1/2}$ were $\mathbf{Z}$ has i.i.d. Gaussian entries (as in Section 5), then $\mathbf{W} = \mathbf{X}^\top\mathbf{X}$ is known as the pseudo-Wishart distribution (also called the singular Wishart), denoted as $\mathbf{W}\sim\mathcal{PW}(\mathbf{\Sigma},n)$, and the variance term from the MSE can be written as $\sigma^2\mathbb{E}[\mathrm{tr}(\mathbf{W}^\dagger)]$. [Sri03] first derived the probability density function of the pseudo-Wishart distribution, and [CF11] computed the first and second moments of generalized inverses. However, for the Moore-Penrose inverse and arbitrary covariance

$\mathbf{\Sigma}$, [CF11] claims that the quantities required to express the mean "do not have tractable closed-form representation." The bias term, $\mathbf{w}^{*\top}\mathbb{E}[\mathbf{I} - \mathbf{X}^{\dagger}\mathbf{X}]\mathbf{w}^{*}$, has connections to directional statistics. Using the SVD, we have the equivalent representation $\mathbf{X}^{\dagger}\mathbf{X} = \mathbf{V}\mathbf{V}^{\top}$ where $\mathbf{V}$ is an element of the Stiefel manifold $V_{n,d}$ (i.e., orthonormal $n$-frames in $\mathbb{R}^{d}$). The distribution of $\mathbf{V}$ is known as the matrix angular central Gaussian (MACG) distribution [Chi90]. While prior work has considered high dimensional limit theorems [Chi91] as well as density estimation and hypothesis testing [Chi98] on $V_{n,d}$, they only analyzed the invariant measure (which corresponds in our setting to $\mathbf{\Sigma} = \mathbf{I}$), and to our knowledge a closed form expression of $\mathbb{E}[\mathbf{V}\mathbf{V}^{\top}]$ where $\mathbf{V}$ is distributed according to MACG with arbitrary $\mathbf{\Sigma}$ remains an open question.

For analyzing the rate of decay of variance and bias discrepancies (as defined in Section 5), it suffices to only consider diagonal covariance matrices $\mathbf{\Sigma}$. This is because if $\mathbf{\Sigma} = \mathbf{Q}\mathbf{D}\mathbf{Q}^{\top}$ is its eigendecomposition and $\mathbf{X} \sim \mathcal{N}_{n,d}(\mathbf{0}, \mathbf{I}_{n} \otimes \mathbf{Q}\mathbf{D}\mathbf{Q}^{\top})$, then we have for $\mathbf{W} \sim \mathcal{PW}(\mathbf{\Sigma}, n)$ that $\mathbf{W} \overset{d}{=} \mathbf{X}^{\top}\mathbf{X}$ and hence, defining $\widetilde{\mathbf{X}} \sim \mathcal{N}_{n,d}(\mathbf{0}, \mathbf{I}_{n} \otimes \mathbf{D})$, by linearity and unitary invariance of trace,

$$\mathbb{E}[\operatorname{tr}(\mathbf{W}^{\dagger})] = \operatorname{tr}\big(\mathbb{E}[(\mathbf{X}^{\top}\mathbf{X})^{\dagger}]\big) = \operatorname{tr}\Big(\mathbf{Q}\mathbb{E}\big[(\widetilde{\mathbf{X}}^{\top}\widetilde{\mathbf{X}})^{\dagger}\big]\mathbf{Q}^{\top}\Big) = \operatorname{tr}\Big(\mathbb{E}\big[(\widetilde{\mathbf{X}}^{\top}\widetilde{\mathbf{X}})^{\dagger}\big]\Big) = \mathbb{E}\Big[\operatorname{tr}\big((\widetilde{\mathbf{X}}^{\top}\widetilde{\mathbf{X}})^{\dagger}\big)\Big].$$

Similarly, we have that $\mathbb{E}[\mathbf{X}^{\dagger}\mathbf{X}] = \mathbf{Q}\mathbb{E}\big[\widetilde{\mathbf{X}}^{\dagger}\widetilde{\mathbf{X}}\big]\mathbf{Q}^{\top}$, and a simple calculation shows that the bias discrepancy is also independent of the choice of matrix $\mathbf{Q}$.

In our experiments, we increase $d$ while keeping the aspect ratio $n/d$ fixed and examining the rate of decay of the discrepancies. We estimate $\mathbb{E}\big[\operatorname{tr}(\mathbf{W}^{\dagger})\big]$ (for the variance) and $\mathbb{E}[\mathbf{I} - \mathbf{X}^{\dagger}\mathbf{X}]$ (for the bias) through Monte Carlo sampling. Confidence intervals are constructed using ordinary bootstrapping for the variance. We rewrite the supremum over $\mathbf{w}$ in bias discrepancy as a spectral norm:

$$\big\|\mathcal{B}(\mathbf{\Sigma}, n)^{-\frac{1}{2}}\mathbb{E}[\mathbf{I} - \mathbf{X}^{\dagger}\mathbf{X}]\mathcal{B}(\mathbf{\Sigma}, n)^{-\frac{1}{2}} - \mathbf{I}\big\|,$$

and apply existing methods for constructing bootstrapped operator norm confidence intervals described in [LEM19]. To ensure that estimation noise is sufficiently small, we continually increase the number of Monte Carlo samples until the bootstrap confidence intervals are within $\pm 12.5\%$ of the measured discrepancies. We found that while variance discrepancy required a relatively small number of trials (up to one thousand), estimation noise was much larger for the bias discrepancy, and it necessitated over two million trials to obtain good estimates near $d = 100$.

## F.1 Eigenvalue decay profiles

Letting $\lambda_{i}(\mathbf{\Sigma})$ be the $i$th largest eigenvalue of $\mathbf{\Sigma}$, we consider the following eigenvalue profiles (visualized in Figure 3):

- `diag_linear`: linear decay, $\lambda_{i}(\mathbf{\Sigma}) = b - ai$;
- `diag_exp`: exponential decay, $\lambda_{i}(\mathbf{\Sigma}) = b\,10^{-ai}$;
- `diag_poly`: fixed-degree polynomial decay, $\lambda_{i}(\mathbf{\Sigma}) = (b - ai)^{2}$;
- `diag_poly_2`: variable-degree polynomial decay, $\lambda_{i}(\mathbf{\Sigma}) = bi^{-a}$.

The constants $a$ and $b$ are chosen to ensure $\lambda_{\max}(\mathbf{\Sigma}) = 1$ and $\lambda_{\min}(\mathbf{\Sigma}) = 10^{-4}$ (i.e., the condition number $\kappa(\mathbf{\Sigma}) = 10^{4}$ remains constant).