[Reviews · NeurIPS 2020]

Review 1

Summary and Contributions: This paper studies the problem of double descent for linear regression models. It uses a random design matrix in which the rows are not independent, but are sampled instead from a determinantal point process. This leads to exact expressions for the risk as well as some exact expression for implicit regularization.

Strengths: The paper deals with a relevant and timely topic, and it is very well written. The significance is in illustrating how a simple change in the design matrix can lead to exact expressions. The claims are illustrated with empirical evaluations.

Weaknesses: I do not see any big weakness in this work.

Correctness: The methodology seems correct

Clarity: Yes, the presentation is very clear and the notation easy to follow.

Relation to Prior Work: The discussion of previous works is very thorough.

Reproducibility: Yes

Additional Feedback:


Review 2

Summary and Contributions: This paper proposes closed-form expressions for the mean squared error of least square regression (Moore-Penrose version), when the data comes is sampled from a determinantal point process. It is also proved that this DPP can be a surrogate model for least squares when the data is i.i.d. sampled from a sub-Gaussian distribution (which is more common). Indeed, the difference between the two MSE goes to zero when both the sample size and the dimension go to infinity in a fixed ratio. As a consequence, the paper recovers and improves known results in the double descent literature. In the process, a new mathematical concept of its own interest is introduced: determinant-preserving random matrices.

Strengths: The result is novel.The assumptions are minimal: general position is virtually always satisfied if the distribution has a density, and homoscedastic is not very demanding. I like the idea of surrogate model: using a random model for X for which we can compute more than in the i.i.d. case is appealing.

Weaknesses: *after rebuttal:* Thanks for the response, I will keep my score as is. --------------------------------------------------------------------------------------------------------- I think this is a very good paper, there are no weaknesses so to speak.

Correctness: As far as I can tell, the proofs are correct. Something that does not seem accurate is the discussion about HMRT19, lines 169-174. Unless I missed something, Figure 2 in the present paper is a plot of the mean squared error MSE=\expec{\norm{\what-\wstar}^2}, as defined line 67. Whereas Figure 2 in HMRT19 is a plot of the generalization risk R=\expec{\norm{X\what-X\wstar}^2}. While these two quantities are related, I fail to see how Figure 2 would contradict the claims in HMRT19.

Clarity: The paper is very well written. typo line 29: thershold \kappa is not defined in Figure 2

Relation to Prior Work: The prior work is discussed extensively.

Reproducibility: Yes

Additional Feedback: Is there any hope that this surrogate design can be used in other situations? Or is this really specific to least squares regression with Moore-Penrose formulation?


Review 3

Summary and Contributions: This paper is concerned with the double descent phenomenon. Therefore a linear regression is studied. The goal of the paper is to calculate the mean squared error (MSE) in a non-asymptotic regime. This is achieved by constructing surrogate random design using the theory of determinantal point processes (DPP). The advantage of this surrogate random design is that the MSE can be calculated exactly. It is shown that the MSE of the original model converges asymptotically with probability one to the MSE of the surrogate random design. ------------------------------------------- After reading the rebuttal and the having the discussion with the other referees, I will keep my score. This is a good paper. One minor issue regarding the broader impact statement: First of all, I want to thank the authors for clarifying that the result, which I cited, does not imply that in unregularized linear regression more data leads to better performance. However, in practice one can assume that one regularizes, potentially making this observation more of an (very interesting!) theoretical phenomenon and maybe less relevant in practice. In my opinion, in the broader impact statement one should refrain from making such rather controversial statements.

Strengths: Recently, the double descent phenomenon has attracted a lot of attention in the ML community. Computing the MSE with high probability non-asymptotically is an important problem. Although this problem is just partially resolved (Conjecture 1) is still open, I think that this paper makes an important step. Also proving the connection with DPPs is interesting.

Weaknesses: -In my opinion, Theorem 3 is the main result in this paper. I think the message of the paper would be much clearer, if Theorem 1 would be put before Theorem 1 and 2. (Theorem 1 is a tool to prove Theorem 3 and its importance becomes only clear through this connection. -Obviously, the paper would be much stronger if Conjecture 1 would have been proven.

Correctness: The proofs seem to be sound.

Clarity: The clarity of writing could be improved: -The main theorems could be stated more clearly. In particular, the authors could state the assumptions more precisely. For example, in Theorem 1 it could be mentioned how $w$ is chosen. (I guess the authors mean an arbitrary $w$.) Since people do not read papers from beginning to end, this would increase the readability a lot.

Relation to Prior Work: Prior work is appropriately described.

Reproducibility: Yes

Additional Feedback: -typo l. 29: "interpolation threshold" l. 198: "product of non-zero eigenvalues". It would be great if the authors could clarify how eigenvalues with algebraic multiplicity larger than one are counted. -Broader impact statement: "Our research can be applied here to provide a theoretical understanding of the surprising phenomenon where more data leads to worse generalization performance." I do not see where this statement is supported in the paper. It also seems to contradict the (recent) paper [1]. It would be great if the authors either explain it better or remove it from the broader impact statement. [1]: arxiv.org/abs/2003.01897 Optimal Regularization Can Mitigate Double Descent Preetum Nakkiran, Prayaag Venkat, Sham Kakade, Tengyu Ma

[Author Response · NeurIPS 2020]

We thank the reviewers for their time and feedback, and will address all documented typos. In addition, we will clarify: (1) our usage of the condition number $\kappa$ in Figure 2, (2) algebraic multiplicity of eigenvalues in the definition of pseudo-determinant (pdet), and (3) that $w$ in Theorem 1 may be arbitrary.

**To Reviewer #2** We agree that our results on MSE (mean-squared error) do not directly contradict the conclusions of Hastie et al. (2019) (specifically section 3.3 result 2) on the generalization risk, which are stated in terms of the mean-squared *prediction* error. We merely observe that our MSE expressions demonstrate that the minimum norm solution has a capacity for learning (in the sense of achieving MSE below that of the null estimator) even when the signal-to-noise ratio is 1. Thanks for pointing this out, we will clarify it further in the final version.

Regarding other applications of surrogate designs, our methodology should be useful whenever the inverse of a random matrix is being studied (e.g., for analyzing randomized Newton-type methods). It is also possible that other surrogate designs can be devised for analyzing other functions of random matrices.

**To Reviewer #3** We agree that an answer to Conjecture 1 would be a strong contribution and are currently actively researching this. However, we believe Theorem 1 should be stated first because we believe it is the more important "exact expression" result which both requires less assumptions and is our starting point for investigating Conjecture 1.

With respect to Reviewer #3's comments on our broader impact statement, our results do not contradict Nakkiran et al. (2020) because they consider regularized i.i.d. designs whereas we consider unregularized i.i.d. / surrogate designs. To improve clarity, we will revise our broader impact statement to (1) make clear that the conclusion arises from the increasing MSE as $n$ increases (in underdetermined $d/n > 1$ regime) and that (2) more data **may** lead to worse generalization, consistent with their earlier findings (Nakkiran et al., 2019) and result 1 of "Towards a more general characterization" subsection.

## References

Hastie, T., Montanari, A., Rosset, S., and Tibshirani, R. J. (2019). Surprises in high-dimensional ridgeless least squares interpolation. *arXiv preprint arXiv:1903.08560*.

Nakkiran, P., Kaplun, G., Bansal, Y., Yang, T., Barak, B., and Sutskever, I. (2019). Deep double descent: Where bigger models and more data hurt. *arXiv preprint arXiv:1912.02292*.

Nakkiran, P., Venkat, P., Kakade, S., and Ma, T. (2020). Optimal regularization can mitigate double descent. *arXiv preprint arXiv:2003.01897*.


[Meta-Review · NeurIPS 2020]

Three knowledgeable referees recommend this paper to be accepted, considering it relevant, timely, and well written. I also recommend accept.